# Non-crossover gene conversions show strong GC bias and unexpected clustering in humans

Amy L Williams[1,2,3†]*, Giulio Genovese[3], Thomas Dyer[4], Nicolas Altemose[5], Katherine Truax[4], Goo Jun[6], Nick Patterson[3], Simon R Myers[5], Joanne E Curran[4], Ravi Duggirala[4], John Blangero[4], David Reich[3,7,8], Molly Przeworski[1,2] on behalf of the T2D-GENES Consortium

[1]Department of Biological Sciences, Columbia University, New York, United States; [2]Department of Systems Biology, Columbia University, New York, United States; [3]Program in Medical and Population Genetics, Broad Institute of Harvard and MIT, Cambridge, United States; [4]Department of Genetics, Texas Biomedical Research Institute, San Antonio, United States; [5]Wellcome Trust Centre for Human Genetics, Oxford University, Oxford, United Kingdom; [6]Department of Biostatistics, University of Michigan, Ann Arbor, United States; [7]Department of Genetics, Harvard Medical School, Boston, United States; [8]Howard Hughes Medical Institute, Harvard Medical School, Boston, United States

*For correspondence: alw289@cornell.edu

Present address: †Department of Biological Statistics and Computational Biology, Cornell University, Ithaca, United States

**Abstract** Although the past decade has seen tremendous progress in our understanding of fine-scale recombination, little is known about non-crossover (NCO) gene conversion. We report the first genome-wide study of NCO events in humans. Using SNP array data from 98 meioses, we identified 103 sites affected by NCO, of which 50/52 were confirmed in sequence data. Overlap with double strand break (DSB) hotspots indicates that most of the events are likely of meiotic origin. We estimate that a site is involved in a NCO at a rate of $5.9 \times 10^{-6}$/bp/generation, consistent with sperm-typing studies, and infer that tract lengths span at least an order of magnitude. Observed NCO events show strong allelic bias at heterozygous AT/GC SNPs, with 68% (58–78%) transmitting GC alleles (p = $5 \times 10^{-4}$). Strikingly, in 4 of 15 regions with resequencing data, multiple disjoint NCO tracts cluster in close proximity (~20–30 kb), a phenomenon not previously seen in mammals.

## Introduction

Meiotic recombination is a process that deliberately inflicts double strand breaks (DSBs) on the genome, leading to their repair as either crossover (CO) or non-crossover (NCO) resolutions. COs play an essential role in the segregation of chromosomes during meiosis whereas NCOs are thought to aid in homolog pairing or in shaping the distribution of COs over the genome (*Cole et al., 2012b*; *Baudat et al., 2013*). While the past decade has seen tremendous progress in our characterization of DSBs and COs in mammals (*Baudat et al., 2013*), little is known about NCO events.

These two resolutions appear to result from a choice early on in the repair of DSB breaks (*Youds and Boulton, 2011*), with a number of properties differing between them (*Cole et al., 2012b*, *2014*). In particular, both outcomes are accompanied by a short gene conversion tract that fills in the DSB on one homologous chromosome with the sequence from the other homolog. Whereas COs yield chromosomes with multi-megabase long segments from each homolog (*Baudat et al., 2013*), NCO gene conversion tracts have been estimated to span ~50–1000 bp (*Jeffreys and May, 2004*). Although short, these NCO gene conversion tracts affect sequence variation by breaking down

**eLife digest** The genetic information inside our cells is stored in the form of chromosomes, which are carefully packaged strands of DNA. Most human cells contain a pair of each chromosome: one inherited from the mother and another from the father. Typically, when a human cell divides, it duplicates all of its chromosomes and then places one copy of each into the two new cells.

However, a different process—known as 'meiosis'—occurs when a human cell divides to make the cells involved in sexual reproduction (i.e., egg cells in females and sperm cells in males). First, the cell duplicates all of its chromosomes as before, but then it pairs the chromosomes originally from the mother with the equivalent chromosomes from the father. These paired chromosomes then swap sections of DNA. Next, the cell divides, and the resulting cells divide again; this produces four new cells that each contain a single, unique copy of every chromosome.

In the process of swapping sections of DNA between chromosomes, the DNA molecule inside the chromosome is broken and different sections of DNA are then joined together. This can occur by one of two methods: 'crossover events' that produce a final chromosome made up of long sequences from each of the contributing chromosomes; and 'non-crossover events', where only a small section of DNA is swapped between the chromosomes.

Research has tended to focus on DNA breaks and crossover events. Now, Williams et al. have looked at the genetic sequences transmitted by both parents to 49 humans—revealing information about a total of 98 meioses—and scoured them for evidence of non-crossover events. In addition to finding 103 sites where these events occurred, Williams et al. discovered that non-crossover events are more frequent around sites where crossover events also have a higher frequency. This suggests that the mechanism that initiates non-crossover events is shared with crossovers, and that non-crossover events primarily occur during meiosis. Unexpectedly, in some areas non-crossover events were found close to each other in 'clusters', which had not previously been seen in humans.

Non-crossover events will only produce an observable change if the chromosomes involved have differences in the sequence of the DNA section that is swapped between them. The number of such variable genetic positions that non-crossover events affect in a generation is roughly the same number as the number of newly generated random mutations to the DNA sequence in a generation. Examining the DNA sequences transferred during non-crossover events also shows that two different types of DNA bases (cytosine and guanine) are more likely to be transmitted by a non-crossover event than are the other two bases (adenine and thymine). This bias indicates that non-crossover events are an important factor in driving genome evolution. In the future, sequencing the entire genome—the total genetic material—of many different people could provide further insights into non-crossover events in humans.

linkage disequilibrium (LD) within a localized region, and, together with COs, are necessary to explain present-day haplotype diversity (*Przeworski and Wall, 2001*; *Ardlie et al., 2001*; *Frisse et al., 2001*).

Despite the importance of NCOs, the frequency with which they occur in mammals remains uncharacterized. Estimates based on the number of DSBs that occur in meiosis suggest that NCOs are an order of magnitude more frequent than COs (*Baudat and de Massy, 2007*; *Cole et al., 2012a*). In turn, sperm-typing studies and analyses of LD indicate that NCOs occur ~1 to 15 fold more frequently than COs (*Ardlie et al., 2001*; *Frisse et al., 2001*; *Jeffreys and May, 2004*; *Gay et al., 2007*; *Odenthal-Hesse et al., 2014*), with this value varying widely in analyses of individual hotspots (*Jeffreys and May, 2004*; *Odenthal-Hesse et al., 2014*). Furthermore, while COs occur at a higher rate in females than in males and tend to occur in different genomic locations between the sexes (*Kong et al., 2010*), there has been no such comparison for NCO events.

The locations of NCO events with respect to recombination hotspots is of interest more broadly. While NCO events are assumed to occur at the same hotspots for DSBs as COs (*Baudat et al., 2013*), in humans, this has only been demonstrated for a limited number of locations in sperm (*Berg et al., 2011*). Furthermore, by considering events in a single meiosis, sperm-typing studies have identified complex COs in which gene conversions tracts occur near but not contiguous with CO breakpoints (*Webb et al., 2008*). A genome-wide analysis of NCO may therefore reveal further features of recombination.

The impact of NCO events on genome evolution is also in need of quantification. Cross-species analyses have shown that in highly recombining regions, GC content increases over evolutionary time, consistent with an important role for GC-biased gene conversion (gBGC) (*Duret and Galtier, 2009*). Polymorphism data also reveal an effect of recombination, with more AT to GC polymorphisms observed in regions of higher recombination (*Auton et al., 2012*; *Pratto et al., 2014*). Moreover, because gBGC acts analogously to positive selection, its effects on polymorphism and divergence can confound studies of human adaptation (*Galtier and Duret, 2007*). Although one recent sperm-typing study reported two recombination hotspots that exhibit GC-bias in NCO resolutions (*Odenthal-Hesse et al., 2014*), most of the evidence of gBGC in mammals is indirect.

Motivated by these considerations, we carried out a study of NCO gene conversion events in pedigrees—to our knowledge, the first genome-wide assay of de novo NCO gene conversion in mammals. We sought answers to the following questions: (1) Do NCO events localize to the same hotspots as COs? (2) What is the rate at which a site is a part of a NCO tract? This is equivalent to the fraction of the genome affected by NCO in a given meiosis. (3) Are there differences in the NCO rate or localization patterns between males and females? (4) What is the strength of NCO-associated gBGC across the genome? (5) Do NCO gene conversion tracts vary substantially in length? (6) Do complex resolutions occur, with discontinuous tracts within a short distance?

We utilized two different sources of data for our analysis. The primary analysis focused on SNP array data from 34 three-generation pedigrees. These SNP array data provide information from 98 meioses, 49 paternal and 49 maternal, and are informative at 12.1 million sites (markers where we can potentially detect a NCO in a parent-child transmission). We followed up with a secondary analysis of a subset of the identified NCO events using whole genome sequence data.

## Results

We carried out a study of de novo meiotic NCO gene conversion resolutions in humans by analyzing Illumina SNP array data at two SNP densities (660 k and 1M SNP density arrays; see 'Materials and methods'—'Samples and sample selection') from 34 three-generation Mexican American pedigrees (*Mitchell et al., 1996*; *Duggirala et al., 1999*; *Hunt et al., 2005*). The goal was to identify de novo NCO gene conversion events, manifested as one or more adjacent SNPs at which the alleles descend from the opposite haplotype relative to flanking markers (*Figure 1A*). Identifying these NCO events requires phasing of genotypes in the pedigree in order to infer haplotypes and the locations of switches between parental homologs in transmitted haplotypes.

Two features make locating NCO events challenging. The first is the density of informative sites. NCO gene conversions have an estimated mean tract length of 300 bp or less (*Jeffreys and May, 2004*; *Odenthal-Hesse et al., 2014*), but on a SNP array with ~1 million variants, genotyped sites occur on average every 3000 bp. Thus SNP array data will identify only a small subset of NCO events. Moreover, to be informative about NCO events (and recombination in general), a site must be heterozygous in the transmitting parent, so not all assayed positions are informative.

The second challenge arises from erroneous genotype calls. Errors in SNP array data can in principle confound an analysis of NCO because certain classes of errors can mimic these events (e.g., if a child is truly heterozygous but is called homozygous, or if a parent is homozygous but called heterozygous). Our study design minimizes false positive NCO calls by using three-generation pedigrees, as depicted in *Figure 1B*. The approach requires that a putative NCO event identified in a child in the second generation is also transmitted to a grandchild (red arrows in *Figure 1B*). Additionally, the approach validates the genotype of the transmitting parent as heterozygous by requiring that the allele from the alternate haplotype in that parent (i.e., the one that is not transmitted in the putative NCO) be transmitted to at least one child (blue arrow in *Figure 1B*). These requirements exclude the possibility that a segregating deletion will be misinterpreted as a NCO event. Moreover, they guarantee that a false positive NCO event will only be called if there are at least two genotyping errors at a site. Specifically, for a false positive to occur, either the recipient of the NCO and his or her child must be incorrectly typed, or the parent transmitting the putative NCO and the child/children receiving the alternate allele must be in error. This approach decreases the number of events that can be detected since not all sites affected by NCO will be transmitted to a grandchild, but importantly it also greatly reduces the false positive rate. Further details on data quality control measures appear in 'Materials and methods' ('Quality control procedures' and 'Pedigree-specific quality control').

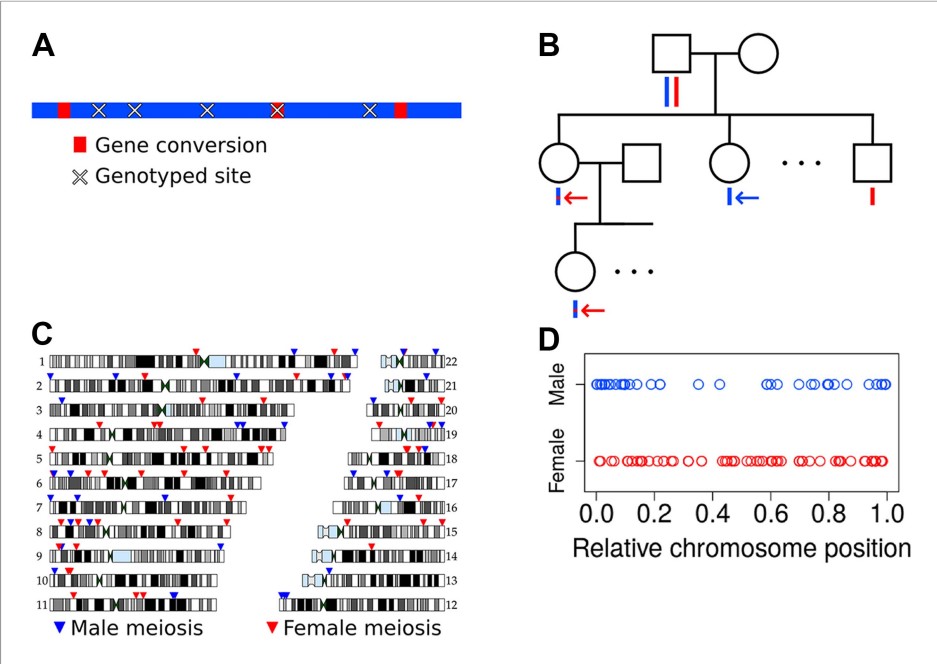

**Figure 1.** Non-crossover detection. (**A**) Pictorial representation of a haplotype transmission including NCO events. A parent has two copies of each chromosome but transmits only one copy to his or her children. That copy is composed of DNA segments from the parent's two homologs; that is, it is formed by recombination between these two haplotypes. Here, the two haplotypes in the parent are colored in blue and red, and switches in color represent sites of recombination. The figure only depicts short NCO events and no COs. Overlaid on this haplotype are × symbols representing sites assayed by the SNP array. In this example, only one NCO has a SNP array site within it and only that NCO can be identified. (**B**) To avoid calling false positive NCO events driven by genotyping error, we required putative NCO events first to be detected in a second generation child (top red arrow) and also transmitted to a third generation grandchild (bottom red arrow). We also required that the allele from the opposite haplotype (i.e., the one not affected by the NCO) in the parent (first generation) be transmitted to at least one child in the second generation (blue arrow). This study design ensures that false positive NCOs will only occur if there are two or more genotyping errors at a site. All 34 pedigrees included in this study have genotype data for both parents, at least three children, one or more grandchild, and both parents of included grandchildren. (**C**) Genomic locations of the NCO sites that we detected are indicated by arrowheads, with red arrowheads representing NCO events from female meioses, and blue from male meioses. Many of the male NCO events localize to the telomeres. (**D**) Relative chromosomal positions of events, stratified by the sex of the transmitting parent.

Our approach for identifying NCO events consisted of, first, phasing each three-generation pedigree using the program HAPI (*Williams et al., 2010*) ('Materials and methods'—'Phasing'). Next, we identified informative sites relative to each parent in the first generation: sites where the parent is heterozygous, the inferred phase is unambiguous, and where, if a NCO event were to occur, both alleles would be transmitted to children (see 'Materials and methods'—'Determination of informative sites'). We then examined all apparent double CO events that occur within a span of 20 informative sites or less, that is, we identified haplotype transmissions that contain switches from one parental haplotype to the other and then switch back to the original haplotype. Most of these recombination intervals span one to three SNPs and are less than 5 kb; these are putative NCO events. A few loci showed complex patterns with multiple, discontinuous recombination events across several SNPs, with tracts spanning 5 kb or more; these are not counted as NCOs but are described further below.

We ascertained the total number of informative sites in the same way as our NCO events. Thus, when calculating the per base pair (bp) rate of NCO, the numerator and denominator are identically ascertained (see below and 'Materials and methods'—'Determination of informative sites' for details).

## Inferred NCOs and their likely source

Within the 34 three-generation pedigrees, we considered transmissions from a total of 98 first generation meioses (49 paternal, 49 maternal). This analysis revealed a total of 103 SNP sites

(henceforth 'NCO sites') putatively affected by autosomal NCO events: 97 with standard ascertainment, and an additional six that are detectable but do not meet all the criteria for inclusion in the rate calculation (*Figure 1C*; *Source code 1*; 'Materials and methods'—'Determination of informative sites'). Most (76/103) NCO events derive from a single SNP, while others contain two or three NCO sites that delimit a tract. The NCO sites have roughly equal numbers of homozygous and heterozygous genotype calls in the recipient (53% heterozygous sites, p = 0.56, two-sided binomial test, this and other statistical analyses included in *Source code 2*), as expected, providing further confidence that the calls are not spurious. Furthermore, we confirmed genotype calls for a subset of the putative NCO events using whole genome sequence data generated by the T2D-GENES Consortium. Sequence data were available for 52 of these NCO sites, of which 50 are concordant with the SNP array calls ('Materials and methods'—'Validating NCO events', *Source code 1*). Of the two discordant sites, one shows evidence of being an artifact in the sequence data rather than the SNP array data, and for the other, the source of error is unclear (see 'Materials and methods'—'Validating NCO events'). Overall, the error rates in these data are low, and so in what follows we assume that all 103 detected NCO events are real.

Meiotic NCOs are thought to localize to the same hotspots as COs (*Baudat et al., 2013*), and studies at specific loci in sperm have supported this hypothesis (*Berg et al., 2011*). To evaluate this question using genome-wide data, we utilized CO rates that Kong et al. estimated based on events identified in an Icelandic pedigree dataset (*Kong et al., 2010*). This genetic map omits telomeres, and thus these rates are only available for a subset of our identified NCOs. The overlapping de novo NCOs show strong enrichment in sites with sex-averaged CO rate ≥10 cM/Mb (*Figure 2—figure supplement 1*). Indeed, 18 of the 72 events that we can examine (26%) localize to such regions (using only one SNP per NCO event), while 4.2% of informative sites have this high a rate. This co-localization is unlikely to occur by chance (p = $8.2 \times 10^{-10}$, one-sided binomial test), indicating that NCOs are strongly enriched in CO hotspots, and providing further validation that the detected NCO events are real.

The enrichment of NCO in regions with high rates of meiotic CO suggests that the NCO resolutions are meiotic in origin. To explore this question further, we compared the locations of the NCO events with a recently reported genome-wide map of meiotic DSB hotspots in human spermatocytes (*Pratto et al., 2014*). We focused our analysis on NCO events transmitted by individuals likely to carry only the *PRDM9* zinc finger A or B alleles (see 'Materials and methods'—'PRDM9 variants'), since individuals with different *PRDM9* zinc finger domains are known to have hotspots in distinct locations (*Baudat et al., 2010*). We further omitted NCO sites that occur near COs (and are consequently ambiguous as to which homolog converted; see below and 'Materials and methods'—'Inclusion criteria'). For this analysis, we considered NCO events rather than single NCO sites, and report an event as overlapping a DSB if any NCO site within it overlaps a DSB. By these criteria, there are 51 events, of which 26 (51%) overlap a meiotic DSB hotspot. Moreover, when focusing on events transmitted by males (because the DSB map is for spermatocytes), 19 of 27 events (70%) overlap a DSB hotspot. This enrichment is highly significant, as only 5.5% of informative sites overlap a DSB hotspot (p < $10^{-8}$, calculated from $10^8$ permutations; see 'Materials and methods'—'Inclusion criteria'). Thus, the NCOs tend to occur at sites of meiotic DSB.

Moreover, the rate at which the NCO events overlap (sex-averaged) historical hotspots inferred from LD is almost identical to the rate at which meiotic DSBs occur in such locations. Considering all unambiguous NCO event locations in male *PRDM9* A/B-only carriers, 56% (15/27) overlap the (population-averaged) LD-based hotspots (*The International HapMap Consortium, 2007*), when between 52% and 63% of DSB hotspots from spermatocytes do, depending on the source population of the LD map analyzed (*Pratto et al., 2014*). The overlap for NCO events from both sexes is similar, with 55% (28/51) of events overlapping LD-based hotspots. Finally, there is no overlap of the NCO events with putative fragile sites (*Fungtammasan et al., 2012*), one of the important sources of mitotic recombination (see *Song et al., 2014*). Given these observations, we conclude that most (possibly all) of our events arose in meiosis.

## Rate of NCO events and their location in the two sexes

The observation of 97 ascertained NCO sites out of 12.1 million informative sites provides an estimate of the rate of NCO per bp. Assuming the set of informative sites is unbiased with respect to the recombination rate, the rate of NCO is equivalent to the number of sites affected by NCO divided by the number of informative sites. This represents the proportion of the genome affected by NCO, or equivalently the probability that a given site will be part of a NCO tract per meiosis.

As *Figure 2A* shows, however, our SNP array data are enriched for regions of high recombination relative to the full genome, and it is necessary to account for this bias. We therefore estimated the rate of NCO in each of six recombination rate intervals based on the HapMap2 recombination map (*Figure 2A*), by dividing the number of NCO sites by the number of informative sites observed in each bin. The overall NCO rate is then the sum of these rates, each weighted by the proportion of the autosomes that occurs in the bin. This procedure yields a sex-averaged rate of $R = 5.9 \times 10^{-6}$ per bp per meiosis (and a 95% confidence interval [CI] of $4.6 \times 10^{-6}$–$7.4 \times 10^{-6}$, calculated by 40,000 bootstrap samples with 10 Mb blocks).

Sperm-typing data have also been used to examine the number and tract length of NCO events. Notably, a study by Jeffreys and May that examined three hotspots in detail (*Jeffreys and May, 2004*) found the number of NCO events to be 4–15 times that of COs, with a mean tract length of 55–290 bp. The rate $R$ can be calculated as the number of NCO tracts in a meiosis multiplied by the tract length and divided by the genome length. Using the estimates from Jeffreys and May yields $R = 2.6 \times 10^{-6}$ to $5.2 \times 10^{-5}$/bp/generation (for a genome-wide CO rate of 1.2 cM/Mb), a range that includes our estimate. Our results are therefore concordant with those from sperm-based analyses; they are also consistent with several LD-based studies of genome-wide levels of NCO (*Ardlie et al., 2001*; *Frisse et al., 2001*; *Gay et al., 2007*).

Considering the parent of origin of each NCO event, we found that the two SNP arrays differ significantly in number of events detected per sex (p = $5.1 \times 10^{-4}$, $\chi^2$ one degree of freedom [df] test), with the lower density SNP dataset uncovering fewer male-specific events than expected. This bias may be caused by a lower coverage of the telomeres in the low density SNP array, and makes the analysis of potential differences in NCO rate between the sexes difficult. Nevertheless, considering the position of events captured by genotype arrays reveals broad-scale localization differences, with male events more prevalent in the telomeres and female events relatively dispersed throughout the genome (*Figure 1C,D*). These sex differences in localization are similar to those seen for CO events (*Kong et al., 2010*), as expected from a shared mechanism for the broader-scale (e.g., megabase-level) control of both types of recombination.

## GC-biased gene conversion

Deviations from the Mendelian expectation of 50% transmission of each allele at a polymorphic site have been observed a number of recombination hotspots in humans. Many of these asymmetries

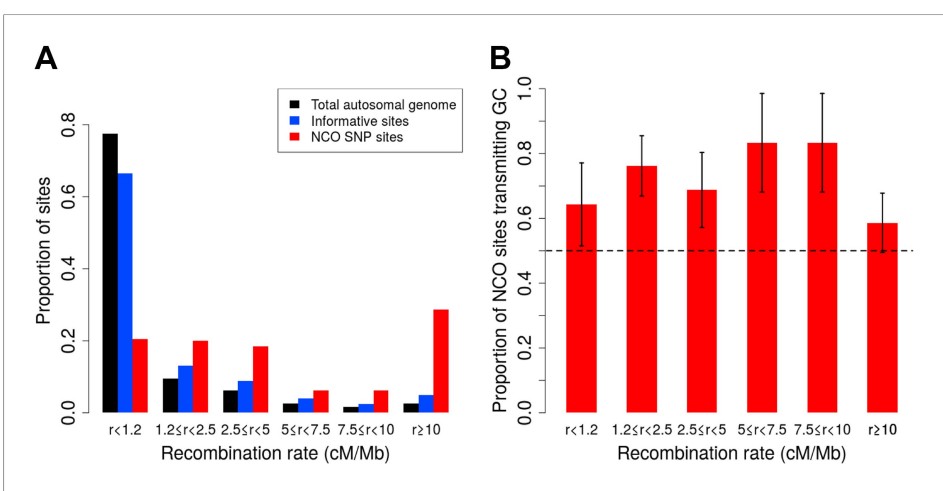

**Figure 2**. Proportion of non-crossover sites and rate of GC vs AT allele transmissions across recombination rate bins. (**A**) Histogram of proportions of sites that fall into six ranges of recombination rates from the HapMap2 LD-based map (*The International HapMap Consortium, 2007*) for the autosomal genome, all informative sites, and the identified NCO sites (see 'Materials and methods'—'Crossover and recombination rates'). (**B**) Rate of transmissions of G or C at AT/GC SNPs, across six recombination rate bins. Plot shows standard error bars.

The following figure supplement is available for figure 2:

**Figure supplement 1**. Proportion of non-crossover sites across crossover rate bins.

result from polymorphisms that occur within motifs bound by PRDM9 (*Webb et al., 2008*). Recombinations at these sites typically show under-transmission of the allele that better matches the *PRDM9* motif, a phenomenon thought to arise through initiation bias due to more frequent breakage of the homolog with a better match to the motif. We identified four NCO events that overlap sequences that match at least six of the eight predictive bps in the degenerate 13-mer motif bound by PRDM9 (*Myers et al., 2008*) (in all four cases, there are exactly 6 of 8 matching bps) and in which a SNP occurs at one of the non-degenerate positions. Because the *PRDM9* motif is GC rich, initiation bias would be expected to predominantly transmit AT alleles, but instead all four of these events transmit GC alleles. Notably however, for three of the events, sequences that match the *PRDM9* motif at seven of eight positions occur at other positions within 2 kb of the NCO site, and for the fourth, another motif with 6 of 8 matching bps occurs within 2 kb. Thus, these four events may not be caused by initiation bias.

A distinct form of bias in transmission that does not depend on the presence of polymorphisms in the *PRDM9* binding motif is thought to occur when AT/GC heteroduplex DNA arises during the resolution of recombination and is preferentially repaired towards GC alleles (*Duret and Galtier, 2009*). A recent sperm-typing study reported on two loci that exhibit such biased gene conversion, associated with NCO but not CO events (*Odenthal-Hesse et al., 2014*). This sperm-based study is, to our knowledge, the first to demonstrate direct evidence of gBGC in mammals. In the NCO events identified here, we saw no evidence for a difference in GC transmission rate between the two SNP density datasets (p = 0.18, $\chi^2$ 1-df test), or between males and females (p = 0.79, $\chi^2$ 1-df test), and so considered the data jointly. For this calculation, we again omitted the ambiguous NCO events (described below and 'Materials and methods'—'Inclusion criteria') and we excluded the four sites that occur within *PRDM9* motifs. The remaining 92 NCO sites all have an AT allele on one homolog and GC on the other, a consequence of the fact that only ≤1% of sites on the Illumina SNP arrays are A/T or C/G SNPs. We observed a strong bias towards the transmission of G or C: of the 92 sites, 63 transmit G or C alleles (68%, 95% CI 58–78%; p = $5.1 \times 10^{-4}$, two-sided binomial test). SNP variants at CpG dinucleotides account for 39 of these 92 sites, and these also show GC bias, with 25 CpG sites (64%) transmitting GC alleles, and no evidence of rate difference between transmissions at CpG and non-CpG sites (p = 0.58, $\chi^2$ 1-df test). By comparison, the sperm-typing study noted above found that two of six assayed hotspots exhibited detectable levels of gBGC, and these two loci transmitted GC alleles in ~70% of NCO transmissions (*Odenthal-Hesse et al., 2014*). Across recombination rate bins, we observed consistent GC transmission rates (p = 0.67, $\chi^2$ 5-df test *Figure 2B*). Since the strength of gBGC depends on both the degree of bias and the rate of recombination, this finding implies that the effects of gBGC will be strongest in high recombination rate regions, as seen in analyses of polymorphism and divergence (*Duret and Galtier, 2009*).

## NCO gene conversion tract lengths

The data allow us to estimate NCO tract lengths, with upper bounds derived from informative SNPs that flank a NCO tract and lower bounds given by the distance spanned by SNPs involved in the same tract. As previously noted, most NCO events involve only one SNP, but a total of twelve regions (ten with information from SNP array data only, and two including information from the sequence data) have tracts that include multiple SNPs (as plotted in *Figure 3*). From these data, we deduced that five of these events have a lower bound on tract length of at least 1 kb while the smallest is at least 94 bp. In turn, one tract is at most 144 bp—only slightly longer than the minimum tract involving more than one SNP (≥94 bp)—and four events have tracts that must be shorter than 1400 bp. These observations, coupled with the variable length in tracts that occur in the clustered NCO events described below (see Figure 4A), suggest that tract lengths span at least an order of magnitude (i.e., 100–1000 bp).

Because NCOs identified using SNP arrays are sparsely sampled, our data may be enriched for events with longer tracts since such tracts impact a larger number of sites. This effect would bias an estimate of the mean tract length using the data from this study. It is also possible that some of the longer events result from clustered but disjoint tracts, as described below. Due to these potential sources of bias, our data cannot be used to learn about mean tract lengths without strong assumptions.

## Complex clustered NCO tracts in sequence and SNP array data

We used Complete Genomics resequencing data generated by the T2D-GENES Consortium to examine variants surrounding several of the identified NCO events at closer resolution. In order to

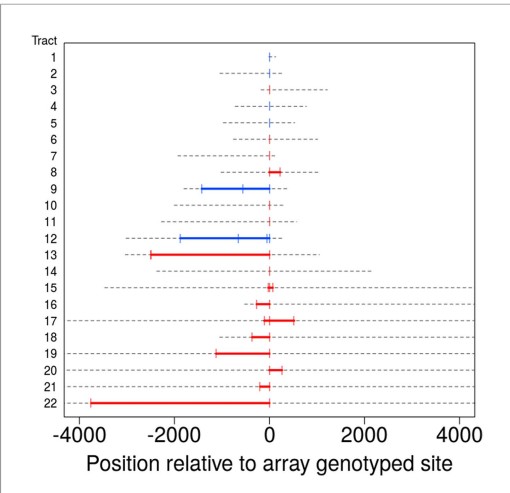

**Figure 3**. Tract lengths for identified non-crossovers. Tract lengths for the 22 NCO events that either have two or more SNPs in a tract or have maximum length of ≤5 kb. Each line corresponds to a NCO tract; lower bounds on length appear in color, with red corresponding to tract lengths informed by SNP array data and blue corresponding to tract lengths from sequence data. Gray dashed lines represent the region of uncertainty surrounding the tract length, with the end points being the upper bound on tract length. Tracts are sorted by the upper bound on tract length.

confidently phase these regions, we required sequence data for both parents and three children (including the NCO event recipient); such data were available for two pedigrees. In these pedigrees, there are a total of 15 regions with evidence for a NCO event in the SNP array data. Two of these regions are not included in this analysis: for one, the sequence data do not contain a genotype call for the site putatively affected by NCO, while in the other, genotype calls do not match the sequence data. Neither locus contains other sites affected by NCO in the sequence data.

*Figure 4A* shows the phase for the 13 regions included. In four cases (haplotypes 10–13), multiple disjoint NCO tracts occur within a short interval of less than 30 kb, with the discontinuities evident from informative sites located between the NCO tracts. Two of these events (haplotypes 11 and 13) occur near COs, and the transmitted haplotypes do not allow us to determine unambiguously which sites experienced the NCO event. (This determination depends on whether the haplotype upstream or downstream of the CO is considered the 'background.') *Figure 4A* plots the NCOs that result in shorter tracts. The four cases occur in a single pedigree, three in the mother, and one in the father (haplotype 11). Using the LD-based genetic map length of the 100 kb around these four regions, we found that this clustering is highly unexpected, with a probability of observing two independent tracts within the four regions ranging from $p = 2.9 \times 10^{-6}$ to $1.9 \times 10^{-4}$ (for each region independently; see 'Materials and methods'—'Examination of regions containing clustered NCOs').

To check for possible artifacts, we performed Sanger sequencing of the three-generation pedigrees for six regions in three of these four haplotypes, indicated by boxes in *Figure 4A*. The Sanger sequence data are concordant with the genotypes from the whole genome sequence data at every site and in all individuals for which we were able to call genotypes (see 'Materials and methods'—'Examination of regions containing clustered NCOs'). We also examined these regions for overlap with segmental duplications, but found none ('Materials and methods'—'Examination of regions containing clustered NCOs'). Finally, to evaluate whether an uncharacterized paralogous sequence variant could confound the results, we considered the genotype status (homozygous or heterozygous) of variants within these regions. Haplotypes 10 and 12 include heterozygous and homozygous genotypes both within and outside the NCO tracts. For haplotypes 11 and 13, the genotypes at all NCO sites are homozygous whereas the genotypes at other sites (blue in *Figure 4A*) are heterozygous. This observation raises the concern of a structural variant or duplicated sequence that has not been identified and spans the nearby CO breakpoint. In this case, heterozygous genotypes could be mismapped to the wrong side of the CO and possibly mimic a NCO tract. Reassuringly, at all these positions, the non-transmitting parent is homozygous, one sibling is heterozygous, the other is homozygous, and neither of the other children received a recombinant haplotype. Thus, all four events appear to comprise true NCOs, but caution is warranted in interpreting two of the four cases.

Intriguingly, the alleles within the NCO events show strong GC bias: For the two unambiguous events (haplotypes 10 and 12), GC alleles were transmitted at 9 out of 10 heterozygous AT/GC SNPs affected by NCO. Moreover, considering all sites in haplotypes 10–13 contained within or between the NCO tracts and irrespective of NCO status, G or C was transmitted at 32 of 43 (74%) heterozygous AT/GC SNPs ('Materials and methods'—'Examination of regions containing clustered

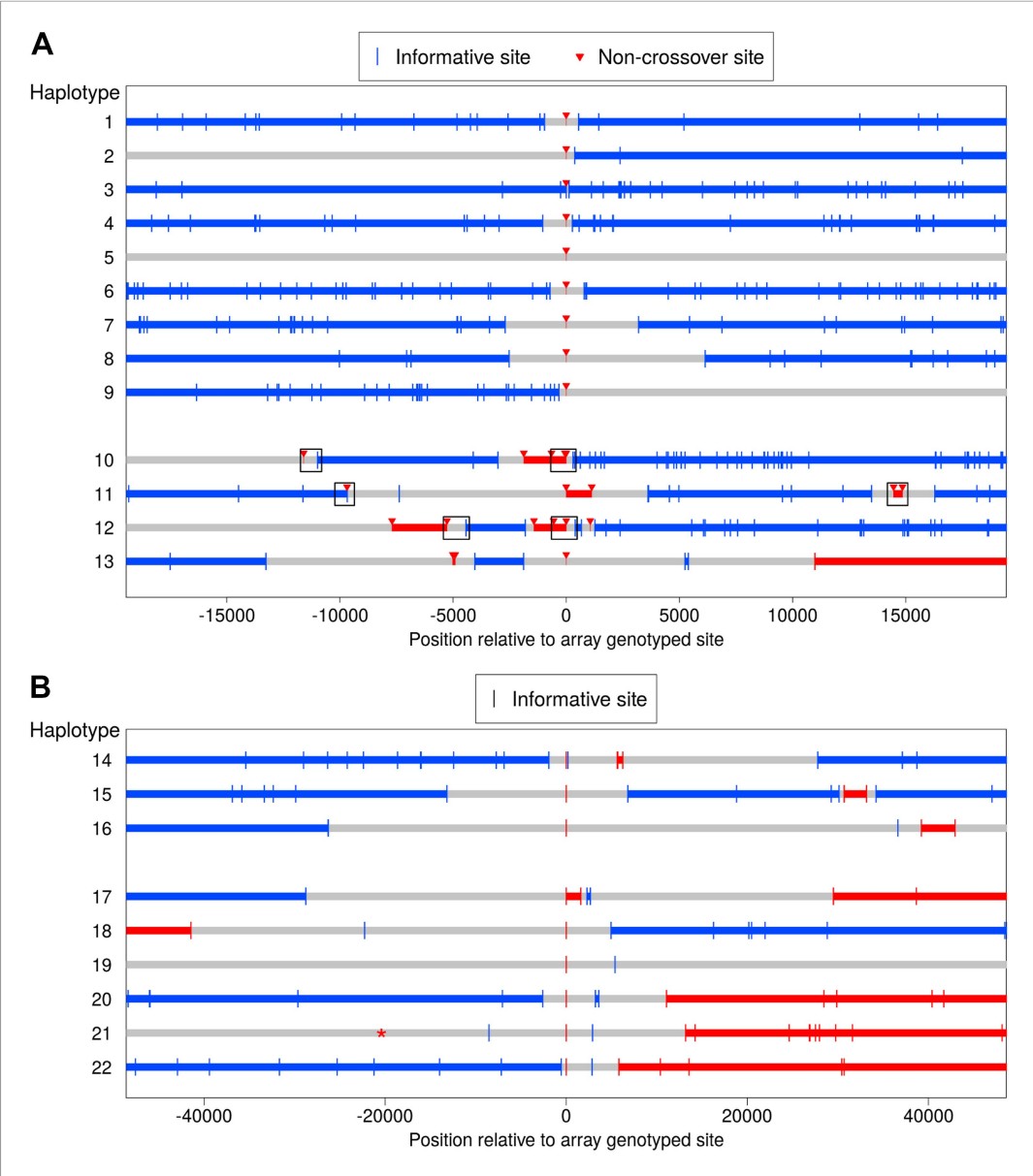

**Figure 4**. Clustered non-crossover events evident in resequencing and SNP array data. (**A**) Recombination patterns in whole genome sequence data for the region surrounding 13 NCO events originally identified in the SNP array data. Each horizontal line represents a haplotype transmission from a single meiosis, and position 0 on the x-axis corresponds to NCO sites identified in the SNP array data. Blue lines depict haplotype segments that derive from the parental homolog transmitted in the wider surrounding region, with blue vertical bars depicting informative sites. Red lines depict segments from the opposite homolog and are putative NCO events, with red arrows indicating informative sites. Grey lines are regions that have ambiguous haplotypic origin. For haplotypes 1–9, only a single site exhibits NCO. For haplotypes 10–13, several NCO sites appear in a short interval near each other but separated by informative SNPs from the background haplotype. Boxes indicate regions for which we preformed Sanger sequencing (see text). (**B**) Clustered recombination events identified in the SNP array data; note the different scale on the x-axis compared with panel **A**. Here, haplotypes 14–16 are clustered NCO events while haplotypes 17–22 occur near but not contiguous with CO events (note the switch in haplotype color between the left and right side of the plot). It is uncertain whether the alleles descending from the blue or the red haplotype represent NCO events ('Materials and methods'—'Inclusion criteria'); thus the plot uses the same symbol for informative sites from both parental haplotypes. Haplotype 19 also appears to have resulted from a CO, but with informative sites more distant than the range of the plot. Haplotype 21 contains an informative marker that has ambiguous phase in the

*Figure 4. continued on next page*

*Figure 4. Continued*

third generation and therefore was not detected initially, but it is plotted here with a * symbol. The ambiguous phase in the third generation is consistent with neighboring sites and not indicative of an incorrect genotype call.

NCOs'). These findings raise the possibility that the patchy repair resolutions observed in the four events result from a GC biased repair process that operates discontinuously within long stretches of heteroduplex DNA. Alternatively, these results could be explained by repeated template switching, as has been observed in *Saccharomyces cerevisiae* (*Tsaponina and Haber James, 2014*).

Further examination of our array-based data revealed additional events: three more clustered NCO events as well as six NCO events near but disjoint from CO resolutions (*Figure 4B*). Two of these haplotypes (numbers 18 and 19) are the same cases that show clustered NCO in sequence data (*Figure 4A*, haplotypes 11 and 13); all other events were seen in distinct pedigrees. These complex CO resolutions shed light on the distances over which such events may occur. The complex CO events previously described in humans were seen in assays of relatively short intervals of ≤4 kb around CO breakpoints, and yielded an estimated frequency of 0.17% (*Webb et al., 2008*). The results from the current study indicate that complex resolutions also occur farther from the CO breakpoint, so may be more common. Whether the observations at short and longer distances result from the same phenomenon remains to be elucidated.

To our knowledge, this is the first observation of clustered but discontinuous NCO gene conversion tracts in mammals, although patterns that resemble those shown in *Figure 4A* have been reported in meiosis (*Martini et al., 2011*; *Globus, 2013*) and mitosis (*St Charles and Petes, 2013*; *Yin and Petes, 2013*) in *S. cerevisiae*. We further note that some events classified as complex CO in humans (based on a limited number of markers) may in fact be complex NCO (*Webb et al., 2008*). The observed complex NCOs and distant forms of complex CO (*Figure 4B*, haplotypes 17–22) both point to a property of mammalian recombination that is poorly understood and in need of further characterization.

## Contiguous and clustered recombination events spanning larger distances

In addition to the NCO events with tracts that span no more than 5 kb, we identified five longer-range recombination events: three continuous tracts, and two that showed a clustering pattern (see *Figure 5*). Each event occurred in a different pedigree; the continuous tract that spans ~79 kb was transmitted by a male, and the four other events occurred in females. The long continuous tracts could conceivably reflect double COs in extremely close proximity, as might arise from a CO-interference independent pathway (*Fledel-Alon et al., 2009*), but the clustered events cannot be explained in this way. For two events, sequence data are available and validate the genotype calls, indicating that the case that spans at least 9 kb in the genotype data is in fact at least 18 kb long (haplotype 23), and confirming the case in which clustered events span ~203 kb (haplotype 26).

Haplotypes 23, 24, and 27 reside on the p arm of chromosome 8 where a long inversion polymorphism occurs (*Antonacci et al., 2009*). Single COs within inversion heterozygotes can be misinterpreted as more than one CO event (*Broman et al., 2003*), yet these three recombination events are >1.7 Mb outside the inversion breakpoints, so should not be affected. One possibility is that the large inversion polymorphism leads to aberrant synapsis during meiosis, leading to complex repair of DSBs. In that regard, we note the transmitter of haplotype 23 is heterozygous for tag SNPs for the 8p23 inversion polymorphism (*Antonacci et al., 2009*), and that a sibling inherited a haplotype from the same parent with a CO at the same position as the end of the tract for haplotype 23. This co-localization may be due to effects of the inversion on synapsis; alternatively, this could indicate that the sites are incorrectly positioned, resulting in inaccurate inference of breakpoint locations (*Broman et al., 2003*). The pattern in haplotype 27 is even more complex and difficult to explain.

## Discussion

NCO gene conversion reshuffles haplotypes and shapes LD patterns, at a rate that we estimate to be $5.9 \times 10^{-6}$/bp/generation. This suggests that roughly 17,110 (95% CI 13,340–21,460) sites will be

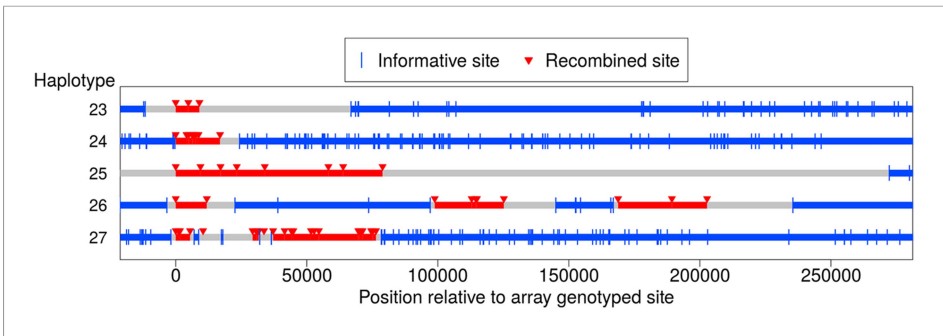

**Figure 5**. Long-range recombination events observed in sequence data. Shown are three contiguous recombination tracts with length ≥ 9 kb, ≥ 16.9 kb, and ≥ 79 kb as well as two sets of clustered long-range recombination events that span ~200 kb and ~76 kb.

affected by NCO in each generation (for a euchromatic genome length of $2.9 \times 10^9$ bp). If the average tract length were 75 bp (consistent with *Jeffreys and May, 2004*; *Cole et al., 2014*), ~228 NCO events (95% CI 178–286) are expected to occur in each generation. Given that the sex-averaged number of COs is ~30 each generation (e.g., [*Fledel-Alon et al., 2011*]), the number of NCOs that we detect is thus in rough agreement with a 10:1 NCO to CO ratio genome-wide (*Baudat and de Massy, 2007*; *Cole et al., 2012a*).

NCO events only impact variation patterns when they occur at heterozygous sites, so in many contexts, this rate is most of interest when scaled by human heterozygosity levels (i.e., the proportion of sites that differ between two homologous chromosomes). Assuming that the heterozygosity rate is $\pi = 10^{-3}$ (*The 1000 Genomes Project Consortium, 2012*), roughly 17 (95% CI 13–21) variable sites are expected to experience NCO in each meiosis. This estimate is on the same order as the number of sites affected by de novo mutation in each generation (*Ségurel et al., 2014*).

In regions that experience NCO, our results indicate that there is frequent over-transmission of G or C alleles. Indeed, we observed GC transmission in 68% of NCO sites (95% CI 59–78%), with no difference in the rate of gBGC across a range of recombination rates (*Figure 2B*). More generally, our results provide a direct confirmation of the presence of gBGC, and lend strong support to the hypothesis that it could play a major role in shaping base composition over evolutionary timescales (*Duret and Galtier, 2009*). Our estimated rate of GC transmission is high relative to what was found in the recent sperm-typing study, where only two of six hotspots had such a bias (~70%) (*Odenthal-Hesse et al., 2014*). In that regard, one possible caveat is that, under certain conditions on mutation, the ascertainment bias of SNP genotyping arrays could lead SNPs subject to stronger biased gene conversion to be enriched, and thus lead us to slightly over-estimate the strength of biased gene conversion across the genome.

Interestingly, a recent reanalysis of data from *S. cerevisiae* (*Mancera et al., 2008*) showed that in this species of yeast, gBGC is associated with CO gene conversions but not NCOs, with a GC transmission rate of ≤55% (*Lesecque et al., 2013*). Both our findings and recent results from human sperm (*Odenthal-Hesse et al., 2014*) indicate that in contrast, in humans, gBGC does operate in NCO events, pointing to a difference in repair mechanisms between humans and yeast that remains to be elucidated.

Considering the distribution of SNPs in NCO tracts, we found lengths that vary over more than an order of magnitude, from hundreds to thousands of base pairs. Intriguingly, we also identified several examples of loci where multiple NCO tracts cluster within 20–30 kb intervals, as well as instances of complex CO over extended intervals. As a potential example of the same phenomenon, a study of de novo mutations reported observing regions with NCO sites across intervals spanning between 2–11 kb (*Campbell et al., 2012*). These events may either be long NCO tracts or clustered but discontinuous NCO events in the same meiosis. In any case, the complex NCO resolutions seen in our pedigree data has not been reported in mammals previously, and is consistent with either with patchy GC biased repair across long stretches of heteroduplex DNA or repeated template switching during the repair of DSBs. Alternatively, these events may arise through mitotic recombination, a process that

has been found to produce similar patterns in yeast (*St Charles and Petes, 2013*; *Yin and Petes, 2013*). Understanding their source will be important for studies of mammalian recombination and for improving population genetic models of haplotypes and LD.

Going forward, whole genome sequencing of human pedigrees will enable unbiased analyses of de novo NCO at relatively high resolution. Of particular interest will be the estimation of the strength of gBGC free of ascertainment bias, as well as systematic examination of tract length distribution and the patterns of complex NCO resolutions revealed by this study.

# Materials and methods

## Samples and sample selection

This study analyzed Mexican American samples from the San Antonio Family Studies (SAFS) pedigrees. SNP array data were generated for these individuals as previously described (*Mitchell et al., 1996*; *Duggirala et al., 1999*; *Hunt et al., 2005*). Our study design required the use of three-generation pedigrees with SNP array data for both parents in the first generation, three or more children in the second generation, one or more grandchildren, and data for both parents for any included grandchildren. Within the entire SAFS dataset of 2490 individuals, there are 35 three-generation pedigrees consisting of 496 individuals that fit these requirements. As noted below, one of these pedigrees was not included in the analysis, so the overall sample consisted of 34 pedigrees and 482 individuals.

Each sample was genotyped using one of the following Illumina arrays: the Human660W, Human1M, Human1M-Duo, or both the HumanHap500 and the HumanExon510S (these latter two arrays together give roughly the same content as the Human1M and Human1M-Duo).

Most of the samples—21 out of the 34 analyzed pedigrees containing 293 individuals—have SNP data derived from arrays with roughly equivalent content and ~1 million genotyped sites. We analyzed all these samples across the SNPs shared among these arrays, with data quality control applied collectively to all samples and sites (see below). After quality control filtering, 896,375 autosomal SNPs remained for the analysis of NCO.

Data for the other 13 out of 34 analyzed pedigrees comprise 189 individuals and were analyzed on a lower density SNP arrays. The majority of the samples in these pedigrees (105 individuals) have SNP array data from ~660,000 genotyped sites. The other samples (84 individuals) have higher density genotype data available, but because other pedigree members have only lower density data, we omit these additional sites from analysis. After quality filtering, this lower SNP density dataset contained 513,283 autosomal sites.

## Quality control procedures applied to full dataset

Initially, sites with non-Mendelian errors, as detected within the entire SAFS pedigree, were set to missing. We next ensured that the locations of the SNPs were correct by aligning SNP probe sequences to the human genome reference (GRCh37) using BWA v0.7.5a-r405 (*Li and Durbin, 2009*). Manifest files for each SNP array list the probe sequences contained on the array and we confirmed that these probe sequences are identical across all arrays for the SNPs shared in common among them. We retained only sites that (a) align to the reference genome with no mismatches at exactly one genomic position and that (b) do not align to any other location with either zero or one mismatches.

We updated the physical positions of the SNPs in accordance with the locations reported by our alignment procedure and utilized SNP rs ids contained in dbSNP at those locations. We omitted sites for which multiple probes aligned to the same location. Some sites had either more than two variants or had non-simple alleles (i.e., not A/C/G/T) reported by dbSNP, and we removed these sites. We also filtered three sites that had differing alleles reported in the raw genotype data as compared to those reported for the corresponding sites in the manifest files. We filtered a small number of sites for which the manifest file listed SNP alleles that differed from those in dbSNP at the aligned location.

Some SNPs are listed in dbSNP as having multiple locations or as 'suspected,' and we removed these sites from our dataset. We also removed sites that occur outside the 'accessible genome' as reported by the 1000 Genomes Project (*The 1000 Genomes Project Consortium, 2012*, roughly 6% of the genome is outside this), and sites that occur in regions that are segmentally duplicated with a Jukes-Cantor K-value of <2% (this value closely approximates divergence between the paralogs; *Bailey et al., 2002*). Finally, we removed sites that occur within a total of 17 Mb of the genome that receive excess read alignment in 1000 Genome Project data (*Genovese et al., 2013*).

We next conducted more standard quality control measures by performing analyses on two distinct datasets: (1) including individuals that were genotyped at ~1 million SNPs (1932 samples) and (2) including all 2490 samples. On the more densely typed dataset, we first removed any site with $\geq$1% missing data and those for which a $\chi^2$ test for differences between male and female allele frequencies showed $|Z| \geq 3$. We then removed 29 samples with $\geq$2% missing data. Next we examined the principal components analysis (PCA) plots (*Patterson et al., 2006*) generated using (a) the genotype data and (b) indicators of missing data at a site. These plots generally show an absence of outlier samples, and the genotype-based PCA plot appears consistent with the admixed history of the Mexican Americans (results not shown).

For the dataset that included samples typed at lower density, we first removed sites with $\geq$1% missing data and sites with male-female allele frequency differences with $|Z| \geq 3$. This filtering step yields SNPs of high quality that are shared across all SNP arrays, including the lower density Human660W array. Next we removed 30 samples with $\geq$2% missing data. Lastly, we examined PCA plots generated using (a) genotype and (b) missing data at each site, and these plots are again generally as expected with an absence of outlier samples (results not shown).

## Phasing and identifying relevant recombination events in three-generation pedigrees

We performed minimum-recombinant phasing on the three-generation pedigrees using the software HAPI (*Williams et al., 2010*), but with minor modifications because this program phases nuclear families independently. Specifically, our approach phased nuclear families starting at the first generation family. After this completed, we phased the families from later generations while utilizing the haplotype assignments from the first generation. Our approach assigned the phase at the first heterozygous marker to be consistent across generations in the individuals shared between the two nuclear families. (Shared individuals are members of the second generation who are a child in one family and a parent in another.) This approach helps produce consistent phasing across generations and does not introduce extra recombinations since the phase assignment at the first marker on a chromosome is arbitrary.

After phasing, our method for detecting NCO events also handled sites with inconsistent phase between the families (though in practice nearly all sites have consistent phase assignments between families). This method excluded individual sites that have inconsistent phase and that occur within a background of flanking markers with consistent phase; we examined these sites individually and confirmed that they do not represent NCO events, but are likely driven by genotyping errors. When 10 or more informative SNPs in succession are inconsistent across families, we assumed that a CO event went undetected in one of the generations, and inverted the phase for the relevant individuals in order to identify putative NCO events.

We analyzed the inferred haplotype transmissions to identify sites that exhibit recombination from one haplotype to the other and then back again. The detection approach identified any recombination events that switch and revert back to the original haplotype within $\leq$20 informative SNPs.

## Pedigree-specific quality control and determination of informative sites

Genotypes are only informative for which haplotype a parent transmits—and therefore recombination—at sites where the parent is heterozygous. We employed a pedigree-specific quality control measure by only considering sites in which all individuals in the full three-generation pedigree have genotype calls and no missing data; other sites are omitted. This requirement helps address possible structural or other complex variants that are specific to a particular pedigree and that may adversely affect genotype calling (as evidenced by a lack of a genotype call for some individual in that pedigree at the given site).

Because NCOs occur relatively infrequently, it is unlikely that the same position will experience NCO in multiple generations. We therefore excluded sites that exhibit NCO in any grandchild (i.e., locations with potential NCO events transmitted from the second generation). We applied this filter regardless of the NCO status in earlier generations in order to obtain unbiased ascertainment of events and informative sites. We also excluded sites that exhibit potential NCO events from a given parent and where that parent only transmits one haplotype. In this case, the genotype from the transmitting parent is likely to be in error and to be homozygous; given this consideration, we considered the site as invalid for both parents.

In principle, all children in the second generation are useful for studying meiosis in their parents, but to reduce false positives, we only analyzed a subset of these children. Specifically, we only considered a child if data for his/her partner and one or more of their children (grandchildren in the larger pedigree) were available.

We counted a site as informative (or not) relative to a given parent and a given child if sufficient data for relatives were available and if it satisfied six requirements. First, we required the parent to be heterozygous at the site. Second, as shown in *Figure 1B*, we required the allele that the given parent transmitted to the child also be transmitted to at least one grandchild. Third, in any series of otherwise informative sites, we counted all but the first and last sites as informative since we detect NCO events as haplotype switches relative to some previous informative site. Fourth, except at sites that are putatively affected by NCO, we required a second child to have received the same haplotype as the child that is potentially informative. This requirement helps to ensure the validity of the heterozygous genotype call of the parent. As an example, consider a pedigree with four children, three of whom received a haplotype 'A' at some site and the fourth of whom received haplotype 'B'. If the fourth child were to receive a NCO at some subsequent position, it would receive haplotype 'A', and thus all four children would receive the same haplotype. This scenario violates the requirement that the alternate allele (not transmitted in the putative NCO) be transmitted to at least one second-generation child. Thus, at this example site, the fourth child is not informative (where it is the sole recipient of haplotype 'B'). Note however that this site could be informative in the other children if they meet the other requirements listed here.

Fifth, we required that the site be phased unambiguously across two generations, and that if a NCO had occurred, the phase at the site would remain unambiguous in the first generation. Sites in which all individuals in a nuclear family are heterozygous have ambiguous phase. Thus, if a given child is homozygous at a marker but all other individuals in the family are heterozygous, the child is not informative at that site since a NCO event would lead the child to be heterozygous. We note that it is possible to identify putative NCOs when a child receives a haplotype that has recombined from otherwise ambiguous phase to be homozygous at this type of marker. Indeed, we identified five such putative NCO sites, but did not include them when calculating the rate of NCO since the denominator does not include ambiguously phased sites and is therefore ascertained differently.

Finally, we imposed further conditions on the transmitted haplotypes and the genotype calls in the third generation. Our focus for these filters was the erroneous case in which a NCO recipient is called homozygous but is truly heterozygous, and his/her partner and children are all heterozygous. Here, the phasing procedure may incorrectly infer that an allele transmitted by the recipient was instead transmitted by his/her partner; thus the potential NCO allele is not necessarily observed in the grandchildren. To address this issue, we filtered sites that have two properties: (1) the recipient is called homozygous, but the partner and grandchildren are all heterozygous, and (2) both parents transmit only one of their haplotypes to the third generation. If, in contrast to property (2), either the recipient or the partner transmits both his/her haplotypes, this type of erroneous NCO genotype call will produce haplotype assignments in the grandchildren with apparent recombination events relative to flanking markers. As a result, there will be an apparent NCO in the third generation and a filter noted above will remove the site from consideration.

## Pedigrees included in the analysis

We excluded one of the 35 available three-generation pedigrees from our analysis. The NCO recipient in this pedigree has a missing data rate that is more than double any other NCO recipient, suggesting genotype quality issues; accordingly, we observed an excessive rate of NCO event calling in this pedigree (results not shown).

## Quality filtering of double recombination events in close proximity

Our method identified all double recombination events (defined as switches from one haplotype to the other and then back again) that span 20 informative sites or fewer. We examined the haplotype transmissions at each such reported event by hand to ensure that segregation to all children and grandchildren matches expectations. A few sites exhibited NCO events in the same interval in two or more children. Because NCO is relatively rare, it is unlikely that these are true events. Additionally, some sites were consistent with NCO events transmitted to the same child from both parents; these

are again unlikely to be real and are more likely caused when a child is homozygous for one allele but called homozygous for the opposite allele. We therefore considered these cases false positives and excluded them from consideration.

Although we omitted sites in which grandchildren exhibit putative NCO events that occur at a single site, the software did not filter putative NCOs that span multiple sites. We examined all events by hand, and excluded three reported NCO events in which the grandchildren either exhibit putative NCOs longer than one SNP (therefore undetected) or show aberrant genotype calls.

The main text describes five long-range recombination events shown in *Figure 5*. For all these events, the recombined alleles at every site were transmitted to the third generation with no apparent recombinations or NCO events in the third generation. We excluded two other events with unexpected transmissions to the grandchildren. Specifically, one 4-SNP contiguous tract shows transmission to the third generation for three of the four recombined SNPs, but one SNP in middle of the tract was not transmitted and shows an apparent NCO in the third generation. The other 18-SNP long contiguous tract shows a putative NCO transmitted from the opposite parent across this same interval. We also excluded an event in which two sites separated by ~27 kb exhibit NCO in the second generation, but where one site has ambiguous phase in the third generation and would not be expected to have such phasing on the basis of flanking markers.

## Validating NCO events

We tested for overrepresentation of either heterozygous or homozygous genotype calls in the recipient of the putative NCOs. Overrepresentation would suggest bias and possibly artifactual detection of NCOs, but we saw no evidence of bias (p = 0.56, two-sided binomial test, this and other statistical analyses included in *Source code 2*). This analysis excludes the five sites identified using non-standard ascertainment and which are homozygous by detection, and also excludes a sixth non-standard site (described below in 'Inclusion criteria').

Of the 482 individuals that we analyzed using SNP array data, 98 were whole genome sequenced by the T2D-GENES Consortium and we were therefore able to check concordance of genotype calls. We attempted validation on all sites for which data were available for the transmitting parent or a recipient (either the child or a grandchild) of the putative NCO site (*Source code 1*). Within these 98 samples, genotype calls were available for 52 of the putative NCO sites (103 total); 42 of these sites include data for both the transmitting parent and a NCO recipient. One additional site had data available for relevant samples, but the sequence data do not contain calls for that position. We compared genotypes for every available parent, child, partner of the NCO recipient, and children of the recipient (grandchildren in the larger pedigree). For ambiguous NCOs, we required data for both possible NCO orientations to be concordant in order to count as validated. The genotype calls for all inspected individuals are concordant between the two sources of data for 50 of the 52 sites. One of the inconsistent sites shows a discordant genotype call between the datasets for the recipient of the NCO, but a concordant call for his child (the grandchild in the pedigree). This inconsistency suggests that the genotype data may in fact be correct. The other discrepancy occurs at a site where sequence data were unavailable for the recipient of the NCO. Here, the genotype call for the transmitting parent is discordant between the two sources of data, and the error source is ambiguous; we retained this site in the analyses.

## CO and recombination rates

CO rates are those reported by deCODE (*Kong et al., 2010*) based on COs detected in large Icelandic pedigrees. The original map is reported for human genome build 36 and was lifted over to build 37 coordinates. This map is estimated to have resolution to roughly 10 kb, and we therefore computed recombination rates in cM/Mb at each site using the genetic distances from the map at the 10 kb surrounding a site and divided by this (10 kb) window size. Because this map omits relatively large telomeric segments, we did not have rates for many sites from the SNP arrays and from the identified NCO events. We used linear interpolation to obtain rates at sites within the range of the map but not directly reported. The proportion of sites in the 'autosomal genome' in *Figure 2—figure supplement 1* derives from all sites within the reported positions in the autosomal genetic map.

The HapMap2 LD-based recombination rates are from the genetic map generated by the HapMap Consortium (*The International HapMap Consortium, 2007*) using LDhat (*McVean et al., 2004*) that was subsequently lifted over to human genome reference GRCh37. We used analogous methods for calculating recombination rates from this map as for the CO map mentioned above, including

a window size of 10 kb and linear interpolation. A few sites on the higher density SNP data (12 of 896,387) fall outside the interval of positions reported in the map and were not included in our analyses.

## PRDM9 variants in the sample

Mexican Americans were previously shown to carry primarily *PRDM9* A and B alleles (*Parvanov et al., 2010*), and admixture with African descent groups may have led to the presence of *PRDM9* C variants. The derived allele at SNP rs6889665 is in strong LD with this *PRDM9* C variant: 96% of haplotypes with the ancestral allele contain <14 zinc fingers, with most being A or B alleles; 93% of haplotypes with the derived allele contain *PRDM9* variants with ≥14 zinc fingers, including primarily the *PRDM9* C variant. With a larger number of zinc fingers, the PRDM9 C variant binds a degenerate 17 bp motif distinct from the motif bound by PRMD9 A and B (*Hinch et al., 2011*). The higher SNP density arrays include genotypes for this site, providing information about the likely *PRDM9* variant of the transmitting parent for 76/103 of the NCO sites. Of these, 11 events are transmitted by a likely *PRDM9* C carrier, with a total of five carrier parents within the 48 parents for which we have genotypes. The remaining 65 events are transmitted by individuals that are homozygous for the ancestral allele at rs6889665 and thus likely to carry only the PRDM9 A or B alleles, both of which bind the common 13-mer motif (*Baudat et al., 2010*).

## Inclusion criteria for NCO and GC-bias rate calculations, hotspots, and tract lengths

Five NCO events were identified with a non-standard ascertainment and are inappropriate for inclusion in estimating the rate of NCO. A sixth, non-standard event is part of a three SNP long tract but has ambiguous phase in the third generation; it appears to be to be a NCO site on the basis of its presence in a tract and the fact that the ambiguous phase in the third generation is consistent with neighboring sites and not suggestive of artifact. None of these sites are expected to show bias with respect to allelic composition and we therefore included them when calculating the strength of GC-bias.

Somewhat more complex cases are NCO sites that occur near CO events (*Figure 4B*, haplotypes 17–22). In most, a single site appears to have been involved in the NCO event, and is followed by a single site that reverts to the first haplotype, and then by a CO. Depending on whether one considers the 'background haplotype' to be the one upstream of the NCO and CO, or downstream of these, the site in the NCO tract differs. Thus the sites affected by NCO are ambiguous. To simplify the examination of GC-bias, we excluded these sites from consideration. The excluded haplotypes are 17–22 (*Figure 4B*); haplotypes 11 and 13 are the same as 18 and 19 and are thus excluded, whereas haplotypes 10 and 12 and 14–16 are unambiguous NCO tracts and are included. (Haplotypes 23–26 in *Figure 5* are long-range events that are not included in any analysis.) Additionally, to avoid confounding biased repair with initiation bias driven PRMD9 binding, we omit four events that overlap partial matches to the *PRDM9* motif as described in 'Results' ('GC-biased gene conversion').

To estimate the rate of NCO genome-wide, rather than exclude the ambiguous NCO sites noted above—which would bias our rate calculation downwards—we instead included both possibilities in the rate calculation, and gave each of them a weight of 0.5, while other sites have a weight of 1. There are two effects of this weighting. First, if the recombination rate bin differs across these sites, they each contribute the weight of half a site to the rate calculation for those bins. Most sites fall into the same rate bin and therefore have the same effect as counting a single site. The second effect of weighting these sites is that, in one case, we cannot tell whether the NCO was two SNPs or only one SNP long. In this case, we counted the event as 1.5 NCO sites. Finally, we observed one instance of two adjacent putative NCO sites separated from a CO by three informative sites. The three informative sites span 19.6 kb—longer than our threshold for NCO events. In this case, we considered the two sites (which form a tract of length at least 264 bp) as part of a definitive NCO with weight 1.

For estimating the number of sites with CO rate ≥10 cM/Mb, we included only one SNP per tract and weighted ambiguous cases by 0.5 as above. Additionally, two ambiguous sites have CO rates that straddle this threshold, with one site slightly less, the other slightly more. To be conservative in estimating a p-value, we considered these sites as falling below the threshold.

We checked overlap between DSB hotspots from *PRDM9* A/A and A/B type individuals determined by *Pratto et al. (2014)* and the set of NCO sites that are unambiguous (i.e., omitting

haplotypes 17–22 from *Figure 4B*) and for which the transmitting parent is likely to carry only *PRDM9* A or B alleles (see 'PRDM9 variants' above). A secondary analysis of these DSB hotspots included only events transmitted by males. To calculate overlap with LD-based hotspots (*The International HapMap Consortium, 2007*), we again included only unambiguous events but did not further restrict the analysis. For both DSB and LD hotspots, we counted overlap with respect to events (some of which include multiple converted SNPs) rather than single NCO SNP sites, and we defined a NCO event as overlapping a hotspot if any of its sites overlap. To assess the significance of the overlap, we performed permutation by randomly sampling NCO sites among all informative sites, selecting the same number of adjacent SNPs as observed for each event. We repeated this process $10^8$ times, each time calculating the proportion of NCO events that overlapped a DSB. Out of $10^8$ permutations, no samples obtained at least the level of overlap seen for the actual NCO events; thus p $< 10^{-8}$.

To examine tract lengths, we omitted all but one ambiguous event. For the one included ambiguous event, the two possibilities have tract lengths ≥1615 bp and ≥365 bp (upper bounds are more than 25 kb for both). We included the shorter of these lengths (365 bp) since this lower bound holds for both possibilities. We note that the addition of the sixth, non-standard NCO site that is part of a three-SNP tract (see above) leads to a minimum tract length of 629 bp instead of 520 bp (obtained for the two-SNP tract identified with standard ascertainment).

## Examination of regions containing clustered NCOs

We calculated the probability of two NCO events occurring within the four intervals in which we observed clustered NCO by rescaling the genetic distances of 100 kb surrounding these regions as reported in the LD-based map. (Note that this map includes some of the historical effects of NCO *McVean et al., 2004*.) We earlier estimated the per bp rate of NCO R, and $R = N \times l/G$ where N is the number of NCO events that occur in a meiosis, l is the average tract length of these events, and G is the total genome length. The genome-wide average rate of *initiation* of NCO at a bp is simply $N/G = R/l$. For an interval with genetic map length d cM, we estimated the rate of initiating a NCO by rescaling this rate as $r = d/c \times R/l$, where c = 1.2 cM/Mb is the average genome-wide rate of CO, and where we assume l = 75 bp. The probability of two independent NCO tracts (conservatively assuming lack of interference among events) is then $P = r^2$. This calculation assumes the HapMap2 map accurately represents the relative rate of both CO and NCO events in an interval; a test for difference between the observed locations of NCO sites and expected locations based on this map are generally consistent with this assumption (p = 0.15, $\chi^2$ 5-df test).

We performed Sanger sequencing on individuals from the three-generation pedigrees in which clustered NCOs occurred. Assayed samples included both parents, all children (including the NCO recipient), the partner of the NCO recipient, and all (four or five) grandchildren of that couple. Overall, sequencing included 11 or 12 samples for each of the three regions examined. We manually examined chromatograms to determine genotype calls. For haplotype 10 (*Figure 4A*), Sanger sequence data overlaps five SNPs called in the Complete Genomics data. For three of these five positions, the sequence data quality was sufficient to easily call genotypes in all samples, whereas for two positions, we called genotypes only in the four grandchildren. Three of the four grandchildren received the haplotype that resulted from a NCO, providing validation of the event at these sites. In all cases, the Sanger-based genotypes are concordant with the Complete Genomics genotypes. For haplotype 11, the Sanger sequence overlapped four SNPs called in the Complete Genomics data. For one of these sites, we called genotypes in all samples, and for two others, we omitted genotypes for one sibling of the recipient but called all other samples. For the fourth site, we could not determine the genotype of the transmitting parent and were uncertain of three of the four siblings of the recipient, but still obtained genotypes for the recipient, one sibling, and four grandchildren. At all four sites, the genotypes that we obtained are consistent with those in the Complete Genomics data. Finally, for haplotype 12, the Sanger sequence overlaps eight sites from the Complete Genomics data. We called genotypes in the five grandchildren at seven of the eight sites, and call four of the five grandchildren at the eighth site. Data quality for other individuals in the pedigree was high for four of the eight sites, but low for the other four sites. In all cases for which we obtained genotype calls, the Sanger data are concordant with the Complete Genomics data. Overall, the Sanger sequence data provided genotype calls for three (haplotypes 11 and 12) or four (haplotype 10) NCO sites (*Figure 4A*) as well as one (haplotype 11) and at least two sites (haplotypes 10 and 12) that descend from the background haplotype (red in *Figure 4A*).

We also checked the regions for potential mismapping from paralogous sequences elsewhere in the genome. Specifically, we looked for overlap between these regions and the following resources: (a) recent segmental duplications that are <2% diverged (*Bailey et al., 2002*; http://ftp.1000genomes.ebi.ac.uk/vol1/ftp/technical/reference/phase2_reference_assembly_sequence/); (b) the 35.4 Mb 'decoy sequences' released by the 1000 Genomes Project (http://ftp.1000genomes.ebi.ac.uk/vol1/ftp/technical/reference/phase2_reference_assembly_sequence/), which contain regions of the genome that are paralogous to sequence from Genbank (*Benson et al., 2014*) and the HuRef alternate genome assembly (*Levy et al., 2007*); and (c) regions of the genome with excess read mapping in the 1000 Genomes Project (*Genovese et al., 2013*). Our quality control procedure already removed individual SNPs that overlap several of these resources ('Materials and methods'—'Quality control procedures'), and this additional analysis revealed no overlap with the regions containing these clustered events.

We examined GC transmission rates for sites within and between NCO tracts in haplotypes 10–13 ('Results'—'Complex clustered NCO tracts'). As previously noted, haplotypes 11 and 13 are ambiguous with respect to NCO status; for these, we included all sites between NCO tracts for both possible recombination outcomes. None of the included sites are definitively on the opposite side of the nearby CO event in the haplotypes.

## Sanger sequencing

We ran Primer3 (http://bioinfo.ut.ee/primer3/) using the initial presets on the human reference sequence from targeted regions to obtain primer sequences. For the suggested primer designs, we performed a BLAST against the human reference to ensure that each primer is unique, and ordered primers from Eurofins Operon. We tested each primer using the temperature suggested during primer design on DNA at a concentration of 10 ng/ul and checked on a 2% agarose gel. For any primer with poor performance, we conducted a temperature gradient, and, if needed, a salt gradient until we found a PCR mix that performed well. Next we performed PCR on the samples of interest, running a small quantity on a 2% agarose gel. We then cleaned the PCR sample using Affymetrix ExoSAP-IT and ran sequencing reactions twice for each sample using Life Technologies BigDye Terminator v3.1 Cycle Sequencing Kit. Finally, we purified each sample using Life Technologies BigDye XTerminator Purification Kit and placed these onto the 3730xl DNA Analyzer for sequencing.

## Acknowledgements

We thank Scott Keeney, Maria Jasin, John Schimenti, Laure Ségurel, and Lorraine Symington for helpful discussions and Melanie Carless for bioinformatics support. We thank Swapan Mallick for sharing a version of the deCODE crossover map in GRCh37 coordinates. ALW was supported by the NIH Ruth L Kirschstein National Research Service Award number F32 HG005944 and by NIH GM83098 to MP. This work was partly completed while MP was a Howard Hughes Medical Institute Early Career Scientist. DR is a Howard Hughes Medical Institute Investigator. T2D-GENES project data generation was supported by NIH grants U01 DK085501, U01 DK085524, U01 DK085526, U01 DK085545, and U01 DK085584.

## Additional information

### Competing interests

MP: Reviewing editor, *eLife*. The other authors declare that no competing interests exist.

### Funding

| Funder | Grant reference | Author |
| --- | --- | --- |
| National Institutes of Health (NIH) | T2D-GENES Consortium | Amy L Williams, Giulio Genovese, Thomas Dyer, Nicolas Altemose, Katherine Truax, Goo Jun, Nick Patterson, Simon R Myers, Joanne E Curran, Ravi Duggirala, John Blangero, David Reich, Molly Przeworski |

| Funder | Grant reference | Author |
|---|---|---|
| National Institutes of Health (NIH) | GM83098 | Molly Przeworski |
| Howard Hughes Medical Institute (HHMI) | Early Career Scientist | Molly Przeworski |
| Howard Hughes Medical Institute (HHMI) | Investigator | David Reich |

The funders had no role in study design, data collection and interpretation, or the decision to submit the work for publication.

### Author contributions
ALW, DR, MP, Conception and design, Analysis and interpretation of data, Drafting or revising the article; GG, NP, SRM, JEC, Analysis and interpretation of data, Drafting or revising the article; TD, KT, GJ, RD, JB, Acquisition of data, Drafting or revising the article; NA, Acquisition of data, Analysis and interpretation of data, Drafting or revising the article

### Ethics
Human subjects: Institutional review board exemption was given for this study from the Broad Institute of Harvard and MIT and the Texas Biomedical Research Institute. The analysis was entirely conducted using anonymous identifiers.

## Additional files

### Supplementary files
• Source code 1. Non-crossover event details. TSV file containing information about each NCO site. Descriptions of each column are listed as comments at the beginning of the file.

• Source code 2. R source code containing statistical analyses of NCO events.

### Major datasets
The following previously published datasets were used:

| Author(s) | Year | Dataset title | Dataset ID and/or URL | Database, license, and accessibility information |
|---|---|---|---|---|
| Hunt KJ, Lehman DM, Arya R, Fowler S, Leach RJ, Göring HH, Almasy L, Blangero J, Dyer TD, Duggirala R, Stern MP | 2005 | Family Investigation of Nephropathy and Diabetes (FIND) Study Genome-Wide Linkage Analyses of Type 2 Diabetes in Mexican Americans: The San Antonio Family Diabetes/Gallbladder Study | http://www.ncbi.nlm.nih.gov/projects/gap/cgi-bin/study.cgi?study_id=phs000333.v1.p1 | Publicly available at the NCBI BioProject database under accession number phs000333.v1.p1. |
| T2D-GENES Consortium | 2013 | T2D-GENES Project 2: San Antonio Mexican American Family Studies | http://www.ncbi.nlm.nih.gov/projects/gap/cgi-bin/study.cgi?study_id=phs000462.v1.p1 | Publicly available at the NCBI BioProject database under accession number phs000462.v1.p1. |

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
