## [Decision Letter]

Thank you for sending your work entitled “Non-crossover gene conversions show strong GC bias and unexpected clustering in humans” for consideration at *eLife*. Your article has been favorably evaluated by Stylianos Antonarakis (Senior editor) and three reviewers, one of whom is a member of our Board of Reviewing Editors. One of the three reviewers, Laurent Duret, has agreed to reveal his identity.

The Reviewing editor and the other reviewers discussed their comments before we reached this decision, and the Reviewing editor has assembled the following comments to help you prepare a revised submission.

Williams et al. report the first genome wide analysis of NCO (non-crossovers) in humans. The authors used an elegant approach to detect these events with high accuracy in human pedigrees. Several observations are interesting: the quantification of NCO events, the chromosomal distribution with an over representation in CO (crossover) hotspots and in subtelomeric regions for NCOs from male meiosis. Two findings are potentially important and novel but the interpretations proposed require further analysis in order to be validated.

First the authors detect a transmission bias towards GC and proposed this as evidence for gBGC. This conclusion requires to exclude the possibility for a bias at the level of initiation and to get support for the meiotic origin of these events. Several situations of initiation bias have been reported in particular by the Jeffrey's lab with transmission bias towards GC or AT among CO and NCOs (for instance Table 1 from Sarbajna et al. 2012).

Second the authors detect long conversion tracts, some of them as clusters and/or with adjacent CO. Based on this observation they conclude that these events are inconsistent with canonical models of double strand repair. However, two problems emerge with this interpretation: first the frequency of these events is not known, second and most importantly these events could be mitotic and actually compatible some models of homologous recombination in mitotic cells and/or during DNA replication (i.e. Break induced replication, see Smith et al., Nature 2007, for instance).

The question of the meiotic/mitotic origin of the events is a general concern in the analysis, and may be in part clarified at least for the category of simple events by further comparative analysis between NCO and CO as presented in Figure 2: The data shows 40% of NCO in regions that have less than 2.5cM/Mb. If CO and NCO activities are correlated at this scale (10Kb), one needs to know more accurately the correlation between CO hotspots activity and NCOs. A better stratification of CO activity is necessary, and may clarify this point and then allow a separate analysis of events in hot regions (thus most likely to be meiotic) or in cold regions for transmission bias. Please also clarify the interpolation (size of intervals used, accuracy) for the many (how many?) sites/events where high resolution CO mapping is not available.

Other comments:

1) In the human genome, about 15% of SNPs correspond to G-C or A-T mutations. It is therefore surprising that among the ∼115 SNPs detected as being involved in a NCO gene conversion event, not a single one corresponds to a G/C or A/T polymorphism. Maybe is this due to the design of the SNP arrays? This should be clarified.

2) 100 sites are computed for GC bias. What has been excluded should be clarified. The Methods refer to haplotypes 17-22, but what about 10-13 and 23-26?

3) Budding yeast is the only other organism where gBGC had been previously quantified (Mancera et al. 2008). The transmission bias in gene conversion events is much weaker in yeast (at most 55:45, depending on how it is measured (Lesecque et al. 2013) compared to what is reported here in humans (70:30). One possible explanation is that the intensity of gBGC in yeast is weakened by interference among neighboring SNPs (Lesecque et al. 2013) (the SNP density in the yeast strain that was analyzed is very high, and the direction of the repair at a given SNP is not independent of that at flanking SNPs). One puzzling observation is that in yeast, gBGC is exclusively associated with CO events (Lesecque et al. 2013). This suggests that the molecular mechanisms that cause gBGC might differ between yeasts and mammals. It might be worth discussing these points.

4) The authors go through considerable lengths to ensure that the detected events are not due to mis-scoring and one would like to know the number of events that were discarded at each step. In addition, the authors could provide a positive control for the entire approach by looking at “uninformative” SNPs, i.e. ones where the parent is homozygous for an allele. What fraction of these homozygous alleles would end up being scored as having undergone gene conversion (i.e. having generated the other allele, either mutationally or by misscoring)?

5) The authors should use the term non-crossover (NCO) or gene conversion without CO when possible.

6) Are PRDM9 motifs detected near NCOs (What are the *PRDM9* alleles in the parents?)?

7) As the authors note the analysis of tract length does not allow drawing a general conclusion because one the limitation of the approach is that detection of events is biased for those more likely to cover at least one SNP, a bias towards longer gene conversion tracts. Thus the interpretation is limited and conclusions such as “tract lengths are highly variable” should be removed and revised. Obviously this comment also applies to complex and large conversion tracts, and it would be very misleading not to emphasize that the sample does not provide any clue about the frequencies of these events with respect to single SNPs for instance. As written, the reader is left with the impression that complex events are a significant proportion (may be 10%) of total!

8) Since complex events have been reported, it is not entirely clear why the authors state that this is the first report of clustered and discontinuous gene conversion tracts? See Figure 4 of Webb at al. 2008; see Mlh1 and Mlh3 deficient mice. Also see Martini et al. 2011, for yeast).

9) The computation of events and the Supplementary Table require several clarifications:

The explanation for the “GC” column is missing in the legend. I understood that it indicates whether the transmitted allele was GC (Y) or AT (N), but this should be clearly stated. Furthermore, as explained in the Methods, there are cases where the direction of conversion is ambiguous. Such cases should be indicated as ‘NA’ in the “GC” column.

I am quite confused by all the blank entries in the “validation” column? I understood from the main text that all events where validated by transmission to grandchild and detection of other allele in sibling?

In the legend, tract means more than one SNPs? Figure 3 indicates 22 gene conversions with more than one SNPs, where are those in the Supplementary Table?

Mislabelling (wrong column) of Tract for rs1540038 and rs1943969.

It would help to indicate haplotypes, to be able to link Supplementary Table and figures.

10) Most gene conversions involve one SNP: how many?

11) In the subsection “Quality filtering of double recombination events in close proximity”, you refer to four long-range recombination events? Which ones?

12) In the subsection “Examination of regions containing clustered gene conversions”, when alluding to “for most variants positions…”, please provide numbers. “Sufficient data were available”, what does sufficient mean?

13) How were chosen the 13 events for Complete Genomics resequencing?

14) Discussion: define Pi.

[Editors' note: further revisions were requested prior to acceptance, as described below.]

Thank you for resubmitting your work entitled “Non-crossover gene conversions show strong GC bias and unexpected clustering in humans” for further consideration at *eLife*. Your revised article has been favorably evaluated by Stylianos Antonarakis (Senior editor), and the original three reviewers, one of whom is a member of the Board of Reviewing Editors. The manuscript has been improved but there are some remaining issues that need to be addressed before acceptance, as outlined below.

The revised paper by Williams et al. provides an improved presentation of the data and more accurate interpretations. However several caveats that the authors acknowledge in the text or in their answers are not clearly stated and this must be revised in order to avoid misinterpretations. Such text improvements should make this work suitable for publication.

1) Overlap NCO/recombination, the question of the meiotic origin of NCO.

One important issue raised which was the mitotic vs meiotic origin of NCO is now strengthened by the analysis of locations of NCO with respect to DSB sites mapped by Pratto et al. This information together with that of LD based hotspot provides good evidence that most of NCOs are of meiotic origin as the authors write in the text. The Abstract should however be revised and written accordingly: “most of the events are likely of meiotic origin”.

The criteria for overlap between LB breakpoints and/or DSB hotspots and NCO events should be explained.

Additional comment: Since 17 among 27 male NCOs overlap with male DSBs whereas 9 female NCOs (among 24) overlap with male DSBs, it suggest that a significant proportion of hotspots have different activities between male and female DSB. Not much is known about male/female difference besides Kong et al. data, who inferred a 85% overlap (thus 15% difference) between male and female CO sites but also Paigen et al. 2008 who detect a large proportion (18 over 28 in a sample analyzed) of intervals with significant difference in activity. These aspects could be discussed.

2) The estimation of gene conversion tract length: the authors have to acknowledge the limitation of their analysis. The last statement of the paragraph “suggest that tract lengths span at least an order of magnitude (i.e., 100-1000 bp)” could be misinterpreted. In fact, in the discussion, the authors use the value of 75bp for average tract length. Thus they should mention that their data gives a range of 100-1000bp but these values provide no indication on the mean value of gene conversion tract length.

3) Complex events: the authors have also revised their interpretation about the origin of complex events. But in order to avoid misunderstanding and overinterpretation they should explicitly mention in the Abstract and Discussion that these events could be of mitotic origin. This observation will be of great interest for the field of genome instability, so it does not affect the data to present this as a potential evidence for mitotic instability.

Other comments:

Fragile sites are one source of mitotic events, certainly not all.

In regard to the subsection “Complex clustered non-crossover tracts in sequence and SNP array data”, formally the complex events in mammalian meiosis were indeed detected as complex CO (Webb AJ, et al. 2008, Svetlanov et al. 2008; Guillon et al., 2005), whether they are complex CO or NCO is actually not known since they were identified by PCR.

Sections on complex and clustered events: focus more the presentation of data, move details (i.e. in the first paragraphs of the subsection “Complex clustered non-crossover tracts in sequence and SNP array data”) to Materials and methods. The problem linked to genotypes (heterozygosity) of haplotypes 11 and 13 is not easy to understand; genotyping of the siblings seem to clarify this issue. Remove unnecessary speculation about frequency of clustering.

Please clarify the following point in the subsection “GC-biased gene conversion”: do you mean that another PRDM9 site is detected away from the NCO?

Figure 4 plots shorter tracts are unclear. Is this panel the result of all 13 resequenced?

In the subsection “Complex clustered non-crossover tracts in sequence and SNP array data”, it is unclear what are these 37 SNPs. Are SNPs from CO events included? This would not make sense.

In the Discussion: Rate of gBGC is said to be high, relative to what? 70% of gBGC by Odenthal et al.?

Figure 2 legend: indicate that the LD map was used for this analysis.

Figure 2—figure supplement 1: indicate that the pedigree map was used for this analysis.

---

## [Author Response]

*Williams et al. report the first genome wide analysis of NCO (non-crossovers) in humans. The authors used an elegant approach to detect these events with high accuracy in human pedigrees. Several observations are interesting: the quantification of NCO events, the chromosomal distribution with an over representation in CO (crossover) hotspots and in subtelomeric regions for NCOs from male meiosis. Two findings are potentially important and novel but the interpretations proposed require further analysis in order to be validated*.

*First the authors detect a transmission bias towards GC and proposed this as evidence for gBGC. This conclusion requires to exclude the possibility for a bias at the level of initiation and to get support for the meiotic origin of these events. Several situations of initiation bias have been reported in particular by the Jeffrey's lab with transmission bias towards GC or AT among CO and NCOs (for instance Table 1 from Sarbajna et al. 2012)*.

We are grateful to the reviewers for their positive comments about this work, and for raising the possibility that initiation bias may explain the GC bias that we observe at non-crossover (NCO) sites. We note that because the most common PRDM9 motif in humans is GC rich, a priori, we would expect initiation bias alone to lead to under-transmission of GC relative to AT, and repair bias via GC-biased gene conversion (gBGC) to lead to over-transmission of GC.

To address this question, we considered NCO sites that overlap the degenerate 13-mer PRDM9 binding motif as reported by Myers et al. 2008. To this end, we defined an overlapping NCO site as occurring in a sequence that match at least six of the motif’s eight non-degenerate bases and focused on cases where the NCO SNP occurs at one of those eight bases. Among NCO sites transmitted by individuals that likely carry only the *PRDM9* A or B alleles (see below regarding PRDM9 variants), four NCO sites overlap such motifs, and in all four, the motif matches 6 of the 8 bases. The NCO events at these four sites all transmit a GC allele, the opposite of what would be expected from initiation bias. However, within 2 kb of all of these sites, there are other sequences that match the PRDM9 motif: three events have sequences with matches to 7/8 bases, and the fourth contains another motif in which 6/8 bases match. If these other matches are the ones bound by PRDM9 rather than the analyzed NCO sites, the observed GC transmissions may not derive from initiation bias but rather from bias in repair. In any case, if we exclude these four sites from consideration (as done in the revised manuscript), we continue to observe strong GC bias, with 63/92 = 68% of sites transmitting GC alleles (*P*=8.1×10^-4^).

*Second the authors detect long conversion tracts, some of them as clusters and/or with adjacent CO. Based on this observation they conclude that these events are inconsistent with canonical models of double strand repair. However, two problems emerge with this interpretation: first the frequency of these events is not known, second and most importantly these events could be mitotic and actually compatible some models of homologous recombination in mitotic cells and/or during DNA replication (i.e. Break induced replication, see Smith et al., Nature 2007, for instance)*.

We thank the reviewers for suggesting that we examine whether our events are mitotic or meiotic in origin. As outlined below, we have now conducted two analyses to address this point, and both suggest that most if not all the NCOs are meiotic in origin (please see the subsection “Inferred non-crossovers and their likely source”). Additionally, we do not see evidence that the NCOs tend to occur in fragile sites, as might be expected if they were due to the repair of mitotic breaks (Song et al. 2014). While we cannot rule out mitotic activity for these four events, our analyses of the events in aggregate strongly suggest that most (potentially all) our events are meiotic. Furthermore, two of these complex events occur somewhat close to crossovers (Figure 4: haplotypes 11 and 13 are the same as 18 and 19, respectively), and may have resulted from a single DSB repair.

As alluded to by the reviewers, the finding of complex resolutions may be consistent with template switching between homolog and sister chromatid, as occurs in break-induced replication (BIR). However, in contrast to BIR, we do not see loss of heterozygosity and only one event is telomeric. We have now added a statement about template switching in the manuscript.

Intriguingly, we found evidence for strong GC bias operating in these events: the transmission of GC alleles occurred at 9 of the 10 AT/GC SNPs subject to NCO in haplotypes 10 and 12. Moreover, transmitted alleles across all AT/GC SNPs in haplotypes 10-13 are GC biased, with GC transmissions at 29 of 37 (78%) of sites. These observations raise the possibility that these events resulted from long stretches of heteroduplex DNA that were repaired by a GC biased process that does not discriminate between homologs at mismatching sites. Based on these considerations, we now note in the manuscript that either template switching or GC biased repair at individual sites within long stretches of heteroduplex DNA could potentially explain these results.

*The question of the meiotic/mitotic origin of the events is a general concern in the analysis, and may be in part clarified at least for the category of simple events by further comparative analysis between NCO and CO as presented in*
Figure 2*: The data shows 40% of NCO in regions that have less than 2.5cM/Mb. If CO and NCO activities are correlated at this scale (10Kb), one needs to know more accurately the correlation between CO hotspots activity and NCOs. A better stratification of CO activity is necessary, and may clarify this point and then allow a separate analysis of events in hot regions (thus most likely to be meiotic) or in cold regions for transmission bias. Please also clarify the interpolation (size of intervals used, accuracy) for the many (how many?) sites/events where high resolution CO mapping is not available*.

To address the question of mitotic vs. meiotic origin of our events, we have added several analyses:

A) We examined the overlap between the locations of our NCO sites and the meiotic DSB hotspots recently reported by Pratto et al. (2014). When restricted to unambiguous events transmitted by individuals likely to carry only PRDM9 A or B variants (see below regarding PRDM9 variants), we observed that 51% (26/51) of NCO events overlap a DSB hotspot. Moreover, when further restricted to events transmitted by males (since Pratto et al. analyzed spermatocytes), 70% (19/27) of NCO events overlap. This enrichment is highly significant (*P*<10^-8^).

B) We considered whether NCO events occur in historical hotspots at comparable rates to DSBs in males. When restricting our focus to unambiguous NCO events transmitted by males that likely carry only PRDM9 A or B, we find very high concordance: 56% of NCO events overlap LD hotspots compared to between 52% and 63% of DSB hotspots, depending on the source population of the LD map. This suggests that the observed NCOs arise the same source as DSBs, and that most if not all our events derive from meiotic recombination.

C) We considered whether any of our NCO occur in fragile sites (Fungtammasan et al. 2012), as might be expected if they result from the repair of mitotic breaks (Song et al. 2014), and found no overlap.

In addition, given that the genome-wide average recombination rate is 1.2 cM/Mb, we added a recombination rate bin of ≤ 1.2 cM/Mb to Figure 2. Only 20% of our events occur at sites with a recombination rate ≤ 1.2 cM/Mb. Because 70% of informative sites lie in this bin, the small number of NCOs in regions of low meiotic recombination is again consistent with a meiotic origin of events.

We note that we moved the plot of event localization with respect to CO rate (based on the deCODE linkage map) to the supplement. While this map is useful in determining whether CO and NCO events have a shared origin, it does not include telomeres, so does not have rates for a sizeable number of the NCO events (Kong et al. 2010).

Lastly, Figure 2 in the revision plots the proportion of GC vs. AT alleles transmitted across several LD-based recombination rate bins; genomic positions with these recombination rates should reflect almost entirely meiotic events, and encompass only 30% of the genome. GC bias is evident among sites with a recombination rate ≥ 2.5 cM/Mb, with 39/58 sites transmitting GC alleles (*P*=0.01), suggesting that at least some of the GC bias activity is meiotic in origin.

*Other comments*:

*1) In the human genome, about 15% of SNPs correspond to G-C or A-T mutations. It is therefore surprising that among the ∼115 SNPs detected as being involved in a NCO gene conversion event, not a single one corresponds to a G/C or A/T polymorphism. Maybe is this due to the design of the SNP arrays? This should be clarified*.

The Illumina SNP arrays that we used for our study contain very few C/G and A/T polymorphisms. For example, the Illumina Human1M contains 1,072,319 SNP probes with only 5,634 C/G or A/T sites (0.5%). Our quality control procedure excluded nearly all of these sites, with only nine remaining in the high density dataset. We have added an explanation of the lack of A/T and C/G SNPs to the text to clarify this point.

*2) 100 sites are computed for GC bias. What has been excluded should be clarified. The Methods refer to haplotypes 17-22, but what about 10-13 and 23-26*?

We have added text to the Methods (“Inclusion criteria” section) to clarify this point. Haplotypes 11 and 13 are the same as 18 and 19 and were excluded from GC bias computations. Haplotypes 10 and 12 are NCO events and not near a CO; they are thus included. Haplotypes 23–26 are long-range events (> 5 kb) that we do not know how to interpret and do not consider in any of the analyses of the NCO events reported.

*3) Budding yeast is the only other organism where gBGC had been previously quantified (Mancera et al. 2008). The transmission bias in gene conversion events is much weaker in yeast (at most 55:45, depending on how it is measured (Lesecque et al. 2013) compared to what is reported here in humans (70:30). One possible explanation is that the intensity of gBGC in yeast is weakened by interference among neighboring SNPs (Lesecque et al. 2013) (the SNP density in the yeast strain that was analyzed is very high, and the direction of the repair at a given SNP is not independent of that at flanking SNPs). One puzzling observation is that in yeast, gBGC is exclusively associated with CO events (Lesecque et al. 2013). This suggests that the molecular mechanisms that cause gBGC might differ between yeasts and mammals. It might be worth discussing these points*.

We agree that the contrast with gBGC in yeast is of interest, and we have added these points and references to Mancera et al. and Lescque et al. to the Discussion. With regard to the exclusivity of gBGC to CO in yeast, we note that Odenthal-Hesse et al. (2014) recently observed strong gBGC that is exclusive to NCO events at two hotspots in human sperm. Together with the results from our study, this provides strong evidence for gBGC operating at NCO events in humans and raises the interesting question of why the repair mechanism differs across species.

We note further that our estimated gBGC rate is high, and we have added a sentence to the discussion addressing the possibility that this rate is upwardly biased by the genotyping ascertainment scheme. (Higher frequency alleles could, in principle, be subject to stronger gBGC and be more likely to be included on a SNP genotyping array.)

*4) The authors go through considerable lengths to ensure that the detected events are not due to mis-scoring and one would like to know the number of events that were discarded at each step. In addition, the authors could provide a positive control for the entire approach by looking at* “*uninformative*” *SNPs, i.e. ones where the parent is homozygous for an allele. What fraction of these homozygous alleles would end up being scored as having undergone gene conversion (i.e. having generated the other allele, either mutationally or by misscoring)*?

We calculated the per bp rate of NCO after applying three broad filters, designed to minimize errors. Below we describe what would happen if we lifted each of these filters in turn.

A) We used the third generation of the pedigree, checking: (1) whether the allele subject to NCO was transmitted to the third generation, (2) whether there is an apparent NCO event in the third generation, and (3) whether that generation has ambiguous phase. With third generation filters removed, we detect a total of 172 single SNP events out of 18.7 million informative sites, corresponding to a NCO rate of 9.2×10^-6^ per bp per generation. In contrast, when these filters are in place, we detect 88 single SNP events out of 12.1 million informative sites, suggesting a rate of 7.3×10^-6^ per bp per generation. (We note that these rate estimates differ from that in the text: we here exclude multi-SNP events, do not correct for ambiguous events near CO sites, and do not correct for the bias in recombination rate at informative sites. See Results “Rate of non-crossover events” and Methods “Inclusion criteria”.)

B) We relaxed the filter relating to missing data, enabling NCO detection at sites where there is an individual in the given pedigree that has missing data. With both this filter and the first filter removed, we detect a total of 204 single SNP events out of 18.8 million informative sites, implying a rate of 1.1×10^-5^ per bp per generation.

C) We relaxed the filter that ensures that at least one child inherits each allele from the transmitting parent. With this and the other two filters removed, we detect a total of 217 single SNP events out of 18.8 million informative sites, for an overall rate of 1.2×10^-5^ per bp per generation.

The reviewers further suggest that considering homozygous sites in the parent might serve as positive controls. We were confused by this point; was this perhaps meant to be negative controls, or as an estimate of the false positive rate? In any case, we note that such sites could not contribute to our gene conversion calls, because any putative conversion or mutation event at homozygous sites would necessarily involve a Mendelian error and so would not be considered in our approach. Thus, while such sites are informative about error rates of the SNP chip or mapping errors, they would not be informative about the error rate of our approach to call NCO events.

*5) The authors should use the term non-crossover (NCO) or gene conversion without CO when possible*.

We agree and have edited the manuscript accordingly.

*6) Are PRDM9 motifs detected near NCOs (What are the* PRDM9 *alleles in the parents?)*?

25.2% of our NCO events are within 2 kb of an exact match, whereas 17.8% of all informative sites are, a weak enrichment. However, we believe this approach to be under-powered, given that the motif is ubiquitous in the human genome and is neither necessary nor sufficient for binding. As an example, 99.8% of SNPs in our high density array data are within 2 kb that of a motif that matches ≥ 6/8 sites. Thus, to address the question of the origin of NCO, we instead focused on the overlap with meiotic DSB and historical recombination rates.

As noted above, in four cases, the NCO sites include a partial match to the motif (see subsection “GC-biased gene conversion”).

With respect to the *PRDM9* variants carried by the parents, we do not have this information for most samples. However, the samples genotyped at higher density include the SNP rs6889665, which is in strong linkage disequilibrium with the PRDM9 C allele in a variety of populations (Hinch et al. Nature 2011). We used this SNP to identify likely PRDM9 C carriers and exclude them from our analysis of the overlap of NCO sites with DSB hotspots. We added the genotype status of this SNP to Supplementary file 1 (previously Supplementary Table).

*7) As the authors note the analysis of tract length does not allow drawing a general conclusion because one the limitation of the approach is that detection of events is biased for those more likely to cover at least one SNP, a bias towards longer gene conversion tracts. Thus the interpretation is limited and conclusions such as* “*tract lengths are highly variable*” *should be removed and revised. Obviously this comment also applies to complex and large conversion tracts, and it would be very misleading not to emphasize that the sample does not provide any clue about the frequencies of these events with respect to single SNPs for instance. As written, the reader is left with the impression that complex events are a significant proportion (may be 10%) of total*!

We agree that it is inappropriate to make claims regarding the variance of the tract length or to suggest knowledge of the rate of complex events. Our claim regarding tract lengths addresses only the span of these events, for which we can estimate a minimum bound (i.e., we know that one event is at most 144 bp and others are at least 1 kb). To make this clearer, we modified the text referenced above to read: “suggest that tract lengths span at least an order of magnitude (i.e., 100-1000 bp).”

Regarding the frequency of clustered events, we similarly modified the text to read: “therefore the rate of clustering is likely to be lower than observed here.” We also removed the sentence that said that these events are unlikely to be rare.

*8) Since complex events have been reported, it is not entirely clear why the authors state that this is the first report of clustered and discontinuous gene conversion tracts? (See*
Figure 4
*of Webb at al. 2008; see Mlh1 and Mlh3 deficient mice. Also see Martini et al. 2011, for yeast)*.

We apologize for this misstatement. The reviewers are correct in noting that Webb et al. observed complex CO events in humans. We believe, however, that clustered NCO events have not been reported previously in mammals. We have therefore modified the referenced sentence to read: “To our knowledge, this is the first observation of clustered but discontinuous NCO gene conversion tracts in mammals, although patterns that resemble those shown in Figure 4 have been reported in meiosis (Tsaponina, 2014, Globus, 2013) and mitosis (Martini, 2011, St Charles, 2013) in *S. cerevisiae*.”

If we are missing any salient references (e.g., we could not find the studies of Mlh1 and Mlh3 deficient mice referred to by the reviewers), we will be happy to include them.

*9) The computation of events and the Supplementary Table require several clarifications*:

*The explanation for the* “*GC*” *column is missing in the legend. I understood that it indicates whether the transmitted allele was GC (Y) or AT (N), but this should be clearly stated. Furthermore, as explained in the Methods (in the subsection headed “Inclusion criteria for gene conversion and GC-bias rate calculations, crossover hotspots, and tract length”) there are cases where the direction of conversion is ambiguous. Such cases should be indicated as* ‘*NA*’ *in the* “*GC*” *column*.

*I am quite confused by all the blank entries in the* “*validation*” *column? I understood from the main text that all events where validated by transmission to grandchild and detection of other allele in sibling*?

*In the legend, tract means more than one SNPs?*
Figure 3
*indicates 22 gene conversions with more than one SNPs, where are those in the Supplementary Table*?

*Mislabelling (wrong column) of Tract for rs1540038 and rs1943969*.

*It would help to indicate haplotypes, to be able to link Supplementary Table and figures*.

We apologize for the missing explanation and for the mislabeling in the original Table. We now provide these data in Supplementary file 1 as a TSV (tab separated value) file, and we also include the R script used to generate the results included in the paper. We believe it is most appropriate to maintain the contents of the “GC” column (now labeled “GC_trans”) as Y/N at all sites for completeness, but with the provided R script, it will be apparent that only columns with a “rate_count” of 1 or 0 (the latter are events not counted in the per bp rate) are used to calculate GC bias. To avoid confusion, we relabeled the former “validation” column to “WGS_validation”. This column contains only information about validation using the Complete Genomics whole genome sequence (WGS) data.

We have added tract numbers to Figure 3 and added a column “fig3_tract” to the table for ease in determining the correspondence. Additionally, as requested, we have added a column “haplo_num” corresponding to the haplotype numbers (from Figure 4) for the corresponding conversion sites.

*10) Most gene conversions involve one SNP: how many*?

We have added the following sentence near the beginning of Results: “Most (76/103) NCO events derive from a single SNP, while others contain two or three NCO sites that form a tract.”

*11) In the subsection “Quality filtering of double recombination events in close proximity”, you refer to four long-range recombination events? Which ones*?

This comment references the events depicted in Figure 5. We have modified the text to read: “The main text describes four long-range recombination events shown in Figure 5.”

*12) In the subsection “Examination of regions containing clustered gene conversions”, when alluding to* “*for most variants positions…*”*, please provide numbers.* “*Sufficient data were available*”*, what does sufficient mean*?

We are sorry for the omission and now provide further detail in the paragraph about the Sanger sequencing results.

*13) How were chosen the 13 events for Complete Genomics resequencing*?

The Complete Genomics resequencing data were generated as part of the T2D-GENES Consortium to study type 2 diabetes in these pedigree samples and were selected for reasons unrelated to the work in this manuscript. While the Consortium performed resequencing for several hundred samples, in order to resolve recombination events, the analysis highlighted in Figure 4 utilized families that had data for both parents, and at least three children. Only two such families were contained in the resequencing data, and these families transmitted a total of 15 NCO events, of which 13 were included in Figure 4 (see main text for the reasons for the two excluded events).

To help clarify this point, we modified the text to read “We used Complete Genomics resequencing data generated by the T2D-GENES Consortium to examine variants surrounding several of the identified NCO events more closely. In order to confidently phase these regions, we required sequence data for both parents and three children (including the NCO event recipient); such data were available for two pedigrees.”

*14) Discussion: define Pi*.

We have modified the text to read: “Assuming that the heterozygosity rate is π = 10^-3^ …” and added a definition, as requested.

[Editors' note: further revisions were requested prior to acceptance, as described below.]

*1) Overlap NCO/recombination, the question of the meiotic origin of NCO*.

*One important issue raised which was the mitotic vs meiotic origin of NCO is now strengthened by the analysis of locations of NCO with respect to DSB sites mapped by Pratto et al. This information together with that of LD based hotspot provides good evidence that most of NCOs are of meiotic origin as the authors write in the text. The Abstract should however be revised and written accordingly:* “*most of the events are likely of meiotic origin*”.

We thank the reviewers for their positive comments on the revised manuscript. We have edited the Abstract to state that “most of the events are likely of meiotic origin.”

*The criteria for overlap between LB breakpoints and/or DSB hotspots and NCO events should be explained*.

The main text now states: “For this [DSB overlap] analysis, we focused on NCO events rather than single NCO sites, and report an event as overlapping a DSB if any NCO site within it overlaps a DSB.” The LD hotspot analysis also uses events rather than single sites. For completeness, we added the following to Methods (“Inclusion criteria”): “For both DSB and LD hotspots, we counted overlap with respect to events (some of which include multiple converted SNPs) rather than single NCO SNP sites, and we defined a NCO event as overlapping a hotspot if any of its sites overlap.”

*Additional comment: Since 17 among 27 male NCOs overlap with male DSBs whereas 9 female NCOs (among 24) overlap with male DSBs, it suggest that a significant proportion of hotspots have different activities between male and female DSB. Not much is known about male/female difference besides Kong et al. data, who inferred a 85% overlap (thus 15% difference) between male and female CO sites but also Paigen et al. 2008 who detect a large proportion (18 over 28 in a sample analyzed) of intervals with significant difference in activity. These aspects could be discussed*.

Establishing that there are differences in hotspot utilization between the sexes is not straight-forward, in that it requires sex differences in broad-scale rates to be properly controlled. Recent work by Campbell et al. suggests that for crossovers, hotspot utilization may be slightly higher in males (http://www.ncbi.nlm.nih.gov/pubmed/25695863). Our NCO data do not help elucidate this issue further, as the difference between the rates of overlap between the two sexes is not significant (*P*=0.095, two-tailed Fisher’s exact test).

*2) The estimation of gene conversion tract length: the authors have to acknowledge the limitation of their analysis. The last statement of the paragraph* “*suggest that tract lengths span at least an order of magnitude (i.e., 100-1000 bp)*” *could be misinterpreted. In fact, in the discussion, the authors use the value of 75bp for average tract length. Thus they should mention that their data gives a range of 100-1000bp but these values provide no indication on the mean value of gene conversion tract length*.

The final paragraph in the tract length section describes the sources of bias in estimating tract length. We have added the following sentence to that paragraph to make clear these concerns regarding mean tract length: “Due to these potential sources of biases, our data cannot be used to learn about mean tract lengths without strong assumptions.”

*3) Complex events: the authors have also revised their interpretation about the origin of complex events. But in order to avoid misunderstanding and overinterpretation they should explicitly mention in the Abstract and Discussion that these events could be of mitotic origin. This observation will be of great interest for the field of genome instability, so it does not affect the data to present this as a potential evidence for mitotic instability*.

Due to space limitations in the Abstract (maximum 150 words), we make this point in the main text of the paper. Specifically, we have added the following text to the Discussion: “Alternatively, these events may arise through mitotic recombination, a process that has been found to produce similar patterns in mouse and yeast.”

*Other comments*:

*Fragile sites are one source of mitotic events, certainly not all*.

We have modified the text to read: “Finally, there is no overlap of the NCO events with putative fragile sites [25], one of the important sources of mitotic recombination (see Song, 2014).”

*In regard to the subsection “Complex clustered non-crossover tracts in sequence and SNP array data”, formally the complex events in mammalian meiosis were indeed detected as complex CO (Webb AJ, et al. 2008, Svetlanov et al. 2008; Guillon et al., 2005), whether they are complex CO or NCO is actually not known since they were identified by PCR*.

We agree that a subset of these events could in fact be complex NCO and have modified the text accordingly.

*Sections on complex and clustered events: focus more the presentation of data, move details (i.e. in the first paragraphs of the subsection “Complex clustered non-crossover tracts in sequence and SNP array data”) to Materials and methods. The problem linked to genotypes (heterozygosity) of haplotypes 11 and 13 is not easy to understand; genotyping of the siblings seem to clarify this issue. Remove unnecessary speculation about frequency of clustering*.

We shortened the lines indicated and moved details to the Methods section, “Examination of regions containing clustered non-crossovers”.

Regarding the genotype status in haplotypes 11 and 13, the siblings provide some reassurance, but they did not receive a CO event in these regions. The potential problem arises when an unknown duplication on both sides of a simple CO leads to heterozygous SNPs that map to physically incorrect locations, thus mimicking a complex recombination. We have modified the text to help clarify this issue: “This observation raises the concern of a structural variant or duplicated sequence that has not been identified and spans the nearby CO breakpoint. In this case, heterozygous genotypes could be mismapped to the wrong side of the CO and possibly mimic a NCO tract.”

We have removed the paragraph regarding the frequency of complex events, as requested.

*Please clarify the following point in the subsection “GC-biased gene conversion”: do you mean that another PRDM9 site is detected away from the NCO*?

Yes, these other PRDM9 motifs are near the NCO site but do not overlap with it. We have rephrased this sentence as: “Notably however, for three of the events, sequences that match the PRDM9 motif at 7 of 8 positions occur at other positions within 2 kb of the NCO site, and for the fourth, another motif with 6 of 8 matching bps occurs within 2 kb.”

Figure 4 plots shorter tracts are unclear. Is this panel the result of all 13 resequenced?

We have rephrased the sentence as follows: “Two of these events (haplotypes 11 and 13) occur near COs, and the transmitted haplotypes do not allow us to determine unambiguously which sites experienced the NCO event. (This determination depends on whether the haplotype upstream or downstream of the CO is considered the ‘background.’) Figure 4 depicts the NCOs that result in shorter tracts.”

*In the subsection “Complex clustered non-crossover tracts in sequence and SNP array data”, it is unclear what are these 37 SNPs. Are SNPs from CO events included? This would not make sense*.

We now define this set of SNPs explicitly: it includes sites internal to the complex NCO region, including SNPs from both possible NCO transmissions for ambiguous events. This definition includes 43 AT/GC SNPs, 32 of which transmit G or C alleles (74%).

*In the Discussion: Rate of gBGC is said to be high, relative to what? 70% of gBGC by Odenthal et al.*?

Odenthal et al. found that two of six examined hotspots have transmission rates of ∼70%, and the others show no discernable gBGC; thus their average rate is lower than ours. We have clarified this point by saying: “Our estimated rate of GC transmission is high relative to what was found in the recent sperm-typing study, where only two of six hotspots had such a bias (∼70%) (Odenthal-Hesse, 2014).”

Figure 2
*legend: indicate that the LD map was used for this analysis*.

We have modified this legend to read: “Histogram of proportions of sites that fall into six ranges of recombination rates from the HapMap2 LD-based map (The International HapMap Consortium, 2007)…”.

Figure 2—figure supplement 1*: indicate that the pedigree map was used for this analysis*.

We modified this legend to read: “Histogram of proportions of sites that fall into six ranges of crossover rates from the deCODE pedigree map (Kong, 2010)…”